# REFA: Reference Free Alignment with Fine-Grained Length Control

**Taneesh Gupta**[*][†]   **Rahul Madhavan**[*][‡]

Xuchao Zhang[†]   Chetan Bansal[†]   Saravan Rajmohan[†]

## Abstract

To mitigate reward hacking from response verbosity, modern preference optimization methods are increasingly adopting length normalization (e.g., SimPO, ORPO, LN-DPO). While effective against this bias, we demonstrate that length normalization itself introduces a failure mode: the **URSLA shortcut**. Here models learn to satisfy the alignment objective by prematurely truncating low-quality responses rather than learning from their semantic content. To address this, we introduce **REFA**, a new alignment framework that proposes probabilistic control on a structural token that controls termination. Our core innovation is a new class of regularizers that operate directly on the probability of the End-of-Sequence (EOS) token, a previously unexploited control lever. This token-level intervention provides a principled solution to the URSLA shortcut, ensuring genuine quality improvements. Furthermore, it unlocks a versatile mechanism for managing the alignment-efficiency tradeoff, enabling practitioners to fine-tune models that adhere to specific token budgets. Empirically, REFA achieves a **60.29%** win rate and a **52.17%** length-controlled win rate on AlpacaEval2 with Llama-3-8B-Instruct, demonstrating the power of our token-level control paradigm.

## 1 Introduction

The twin goals of modern Large Language Model (LLM) research are enhancing alignment with human values and improving computational efficiency for practical deployment. While alignment is advanced through preference optimization techniques like DPO (Rafailov et al., 2024b; Xu et al., 2024b; Wang et al., 2023; Ouyang et al., 2022), efficiency is often pursued via model-centric approaches such as quantization (Dettmers et al., 2023), distillation (Shridhar et al., 2023; Agarwal et al., 2023), and architectural innovations that reduce computational load (e.g., Matryoshka representations (Kusupati et al., 2022; Devvrit et al., 2024)). In this work, we argue for a complementary axis of efficiency: managing the inference-time budget by controlling output token length through post-training alignment. The ability to precisely regulate response verbosity is a critical tool for building cost-effective and responsive LLM-powered systems (Wang et al., 2024b; Erol et al., 2025).

However, achieving such control is complicated by a classic problem: reward hacking. In preference datasets, rewards often correlate spuriously with length, creating an incentive for models to generate longer responses to maximize their score (Gao et al., 2023). To counter this, a growing class of preference optimization methods (e.g., SimPO, ORPO, LN-DPO) incorporates length normalization directly into their objective. Yet, this fix for one form of reward hacking inadvertently creates another. As we formalize in Section 3, this leads to the **URSLA shortcut**: a new failure mode where models learn to satisfy the alignment objective by exploiting a statistical property of language generation to prematurely truncate low-quality responses. This presents a dichotomy: the very mechanism needed to prevent one length-based exploit enables another, making robust length control elusive.

---

[*]Equal contribution.

[†]Microsoft

[‡]Indian Institute of Science (IISc), Bangalore

We argue this challenge stems from the coarse-grained nature of current preference objectives, which not only treat sequence probabilities as opaque scores but are often limited to single preference pairs (Zhong et al., 2024; Zeng et al., 2024). To achieve robust alignment and practical control, we introduce **REFA**, a new alignment framework that proposes probabilistic control on a structural token that governs termination. Our framework is designed to leverage the full spectrum of available data by operating on **sets** of preferred and rejected responses (Gupta et al., 2025). The core innovation is a new class of regularizers that operate directly on the probability of the End-of-Sequence (EOS) token. Viewing generation as a sequential decision-making process (Rafailov et al., 2024a), the EOS token acts as the transition to a "terminal state" in the MDP, making it a uniquely powerful point of intervention. This token-level control provides a unified solution: it directly counteracts the URSLA shortcut and transforms output length into a controllable parameter.

This principled control, achieved at training time, enables REFA to serve as a versatile method for managing the alignment-efficiency tradeoff. The ability to fine-tune a model with specific length constraints is crucial for a range of applications. For instance, systems requiring concise outputs, such as generating summaries (Stiennon et al., 2020) or social media posts, can be optimized for brevity. Conversely, for complex reasoning tasks where detailed explanations are paramount (Wei et al., 2022; Zhang et al., 2024), REFA can encourage more verbose and thorough responses. Most critically, for large-scale deployments, our framework allows practitioners to fine-tune models that adhere to strict token budgets, directly managing inference costs and latency without compromising alignment quality.

In this work, we introduce REFA, a framework that establishes this novel approach of token-level control. Theoretically, we formalize the URSLA shortcut as a fundamental limitation of coarse-grained preference optimization. Algorithmically, we pioneer a new class of EOS probability regularizers that provide fine-grained, bidirectional length control. Empirically, this new approach enables REFA to achieve state-of-the-art results in iterative, on-policy alignment, reaching a **60.29%** win rate on AlpacaEval2 with Llama-3-8B-Instruct. Furthermore, we demonstrate REFA's practical utility through budgeted alignment, showcasing its ability to steer generation towards specific length targets, bridging the gap between alignment quality and deployment efficiency.

We now outline our key contributions in this work:

1. **Theoretical Insight: Formalizing the URSLA Shortcut.** We identify and formalize the *Uncertainty Reduction with Sequence Length Assertion (URSLA)* shortcut, a critical failure mode inherent in reference-free alignment methods that use length normalization. We prove that under the URSLA principle, these models are incentivized to shorten negative responses as a trivial means of satisfying the loss, highlighting a fundamental limitation of purely sequence-level optimization.

2. **Algorithmic Novelty: A New Paradigm of Token-Level Control.** We pioneer a new approach to preference optimization by intervening directly at the token level. We propose a new class of **EOS probability regularizers** that act on the model's termination decision, a previously unexploited control lever. This mechanism provides a principled solution to the URSLA shortcut and establishes a new paradigm for fine-grained control over generative properties.

3. **A Practical Toolkit for Budgeted Alignment.** We demonstrate that our token-level intervention is not just a theoretical fix but also a versatile mechanism for managing the crucial alignment-efficiency tradeoff. We develop and validate a family of regularizers that provide practitioners with explicit, bidirectional control over response length, enabling the fine-tuning of models that adhere to specific inference-time token budgets.

4. **State-of-the-Art Alignment Performance.** We show that by resolving these underlying instabilities, REFA achieves state-of-the-art performance. In challenging iterative and on-policy settings, REFA significantly outperforms strong baselines, achieving a **60.29%** win rate and a **52.17%** length-controlled win rate on AlpacaEval2 with Llama-3-8B-Instruct, validating the effectiveness of our approach.

## 2 Related Work

The alignment of Large Language Models (LLMs) has rapidly evolved from early Reinforcement Learning from Human Feedback (RLHF) techniques (Ouyang et al., 2022) to simpler, more stable Direct Preference Optimization (DPO) methods (Rafailov et al., 2024b). Our work builds upon three key trends that have emerged from this paradigm shift.

**Reference-Free and Multi-Preference Optimization.** To improve simplicity and data efficiency, the field is moving beyond the reference-based, pairwise formulation of DPO. **Reference-free** methods like SimPO (Meng et al., 2024) and ORPO (Hong et al., 2024b) eliminate the need for a fixed reference policy, optimizing the model's log-probabilities directly. Concurrently, **multi-preference** methods like InfoNCA (Chen et al., 2024a) and MPO (Gupta et al., 2025) leverage modern datasets with multiple graded responses per query (Cui et al., 2023), enabling a richer, set-wise contrast that better approximates the true preference landscape. REFA operates at this intersection, proposing a reference-free, multi-preference objective.

**Reference-Free Alignment:** A common component in many alignment techniques, including original DPO, is a fixed reference model representing the policy before fine-tuning. Managing this reference model adds complexity, leading to growing interest in *reference-free* methods (Meng et al., 2024; Yuan et al., 2023; Zhao et al., 2023a). These approaches optimize the policy's likelihoods directly, often leveraging richer feedback signals such as scalar rewards available in datasets like UltraFeedback (Cui et al., 2023). This simplification can make the training pipeline more robust and efficient.

**Length as a Failure Mode for Alignment.** A critical challenge in preference optimization is the tendency for models to exploit response length as a spurious correlate for reward. While the initial problem was a bias towards verbosity (Gao et al., 2023), the standard solution—length normalization—introduces a subtle but severe second-order problem, which we term the URSLA shortcut. Prior works have sought to manage length through sequence-level mechanisms like explicit regularization (Park et al., 2024) or by directly normalizing the objective (Meng et al., 2024; Ahrabian et al., 2024). REFA contributes a new approach by arguing that robust control requires moving beyond the sequence level to intervene directly on the model's termination logic at the token level.

**Positioning REFA.** REFA synthesizes these advancements into a single, coherent framework. It is a reference-free, multi-preference alignment method designed specifically to address the fundamental challenge of length-based reward hacking. Unlike prior works that apply coarse, sequence-level corrections, REFA introduces a novel approach to fine-grained control via an End-of-Sequence (EOS) probability regularizer. By identifying and solving the subtle failure modes of length normalization, REFA not only achieves more robust alignment but also provides a practical framework for managing the inference-time budget, directly connecting the goals of preference alignment and computational efficiency.

We provide full details of our related work in Section B.

## 3 The Challenge: A Subtle Length-Based Shortcut in Reference-Free Alignment

The paradigm of preference optimization for aligning Large Language Models (LLMs) has seen a significant shift from reference-based methods like Direct Preference Optimization (DPO) (Rafailov et al., 2024b) towards simpler, more direct reference-free approaches (Meng et al., 2024; Hong et al., 2024b). While eliminating the reference model reduces complexity and can improve performance, it exposes a new set of challenges related to the inherent statistical properties of language and the dynamics of the optimization process itself. This section first establishes the necessary notation and then identifies a critical, second-order problem that arises from the standard solution to length bias, motivating the need for a more sophisticated approach to length control.

### 3.1 Notation and Preliminaries

Let $\mathcal{D}$ be a preference dataset consisting of tuples $(x, \{y_i, r_i\}_{i=1}^K)$, where $x$ is a query, $\{y_i\}_{i=1}^K$ is a set of $K$ candidate responses, and $r_i \in \mathbb{R}$ is a scalar reward indicating the quality of response $y_i$. Our goal is to fine-tune a policy model $\pi_\theta$ to increase the likelihood of high-reward responses.

The mean reward for a given query is $\bar{r} = \frac{1}{K} \sum_{i=1}^K r_i$. We partition the responses into a *positive set* $Y^+ = \{y_i \mid r_i > \bar{r}\}$ and a *negative set* $Y^- = \{y_i \mid r_i \le \bar{r}\}$. A key quantity in reference-free optimization is the policy's log-probability of a response, $\log \pi_\theta(y|x)$. Since raw log-probabilities are biased towards shorter sequences (as they represent a sum over fewer negative log-probability terms), reference-free methods commonly employ the **length-normalized log-probability**:

$$\overline{\log \pi_\theta}(y \mid x) = \frac{1}{|y|} \log \pi_\theta(y \mid x) = \frac{1}{|y|} \sum_{t=1}^{|y|} \log P_\theta(y_t \mid x, y_{<t}), \tag{1}$$

where $|y|$ is the number of tokens in response $y$. This metric reflects the average per-token confidence of the model.

### 3.2 The Reference-Free Dilemma: Length Normalization and the URSLA Shortcut

Reference-free alignment objectives, such as that of SimPO (Meng et al., 2024), typically aim to maximize the margin between the scores of preferred and dispreferred responses. Using the length-normalized log-probability (Eq. 1) as the score is a necessary first step to prevent the model from trivially favoring longer responses, which are often correlated with higher rewards in preference datasets (Dubois et al., 2024).

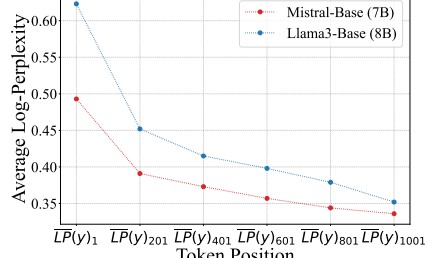

However, this very solution introduces a more subtle and problematic failure mode. The issue stems from a fundamental property of auto-regressive language generation: the relationship between sequence length and model uncertainty. We formalize this relationship in the following conjecture.

**Conjecture 1** (Uncertainty Reduction with Sequence Length Assertion (URSLA)). Let $\overline{LP}(y) = -\overline{\log \pi_\theta}(y \mid x)$ be the average per-token negative log-probability (a measure of model uncertainty or perplexity) for a response $y$. There exists a non-empty subset of coherent responses $\mathcal{Y}' \subseteq \mathcal{Y}$ such that for all $y \in \mathcal{Y}'$, increasing the length of $y$ reduces its average per-token uncertainty. Formally:

Figure 1: Empirical analysis supporting the URSLA conjecture (Conjecture 1) on Ultrafeedback. The plot shows a decreasing trend in average per-token negative log-probability (uncertainty) as sequence length increases for both Mistral-7B and Llama-3-8B base models.

$$\frac{\partial \overline{LP}(y)}{\partial \operatorname{len}(y)} < 0 \quad \text{for all } y \in \mathcal{Y}'. \tag{2}$$

The URSLA conjecture, empirically supported by Figure 1, posits that model uncertainty decreases for longer coherent sequences. This interaction with a length-normalized objective creates a critical vulnerability. Since preference optimization objectives seek to reduce the score $\overline{\log \pi_\theta}(y|x)$ for any negative response $y \in Y^-$, URSLA provides a trivial, non-semantic path to achieve this: by simply reducing the response's length.

This leads to an undesirable optimization shortcut: **the model can satisfy the loss by prematurely terminating rejected sequences, rather than learning the semantic reasons for their low quality.** This reward-hacking behavior undermines genuine alignment, as the model exploits a statistical artifact of length instead of improving its understanding of

quality. Therefore, while length normalization is a prerequisite for reference-free alignment, it is insufficient on its own. It inadvertently creates a new failure mode that must be addressed with an explicit length control mechanism to ensure genuine alignment.

# 4 Developing the REFA Framework

Having established the critical need for explicit length control in reference-free alignment, we now develop the **REFA** (Reference-Free Alignment) framework. Our approach begins by contextualizing REFA within the landscape of existing preference optimization objectives. We then systematically construct a loss function that not only leverages the strengths of prior work but also possesses a theoretically superior alignment objective.

## 4.1 Technical Preliminaries: Key Preference Optimization Objectives

The design of REFA builds upon several key ideas in modern preference optimization. We briefly present the objectives for SimPO, InfoNCA, and MPO, as their components and limitations directly motivate our approach, using the notation established in Section 3.1.

**Simple Preference Optimization (SimPO).** SimPO (Meng et al., 2024) is a prominent reference-free method that operates on preference pairs $(y_w, y_l)$. Its key innovation is using the length-normalized log-probability as an implicit reward signal. The objective maximizes the margin between the scores of the winning and losing responses:

$$L_{\text{SimPO}}(\theta) = -\mathbb{E}_{(x,y_w,y_l)\sim\mathcal{D}} \left[ \log \sigma \left( \beta \overline{\log \pi_\theta}(y_w|x) - \beta \overline{\log \pi_\theta}(y_l|x) - \gamma \right) \right], \tag{3}$$

where $\sigma$ is the sigmoid function and $\gamma$ is a margin hyperparameter. While effective and reference-free, SimPO's primary limitation is its restriction to pairwise comparisons.

**InfoNCA.** InfoNCA (Chen et al., 2024a) addresses the multi-preference setting by leveraging all $K > 2$ responses. It is a reference-based method that minimizes the cross-entropy between a reward-derived target distribution ($p_i^{\text{target}} \propto e^{r_i}$) and a model distribution derived from the policy-reference ratio ($p_i^{\text{model}} \propto \pi_\theta(y_i|x)/\pi_{\text{ref}}(y_i|x)$). Its loss is:

$$L_{\text{InfoNCA}}(\theta; \pi_{\text{ref}}) = -\mathbb{E}_{x\sim\mathcal{D}} \left[ \sum_{i=1}^{K} p_i^{\text{target}} \log p_i^{\text{model}} \right]. \tag{4}$$

InfoNCA's strength is its ability to use all response data, but it requires a reference model and aligns to an explicit target distribution, which, as we will show, can be a suboptimal objective for alignment.

**Multi-Preference Optimization (MPO).** MPO (Gupta et al., 2025) also utilizes $K > 2$ responses but employs a group-contrastive objective. It contrasts the set of positive responses $Y^+$ against all responses $Y$. MPO introduces deviation-based weighting, $w_y = e^{\alpha(r_y - \bar{r})}$, to emphasize more informative examples. Using a reference-based score $s_\theta(y|x) = \log(\pi_\theta(y|x)/\pi_{\text{ref}}(y|x))$, its loss is:

$$L_{\text{MPO}}(\theta; \pi_{\text{ref}}) = -\mathbb{E}_{x\sim\mathcal{D}} \left[ \log \frac{\sum_{y\in Y^+} w_y \exp(\beta s_\theta(y|x))}{\sum_{y\in Y} w_y \exp(\beta s_\theta(y|x))} \right]. \tag{5}$$

MPO's core contributions are its group-contrastive formulation and deviation weighting, though its original formulation retains a reference model. REFA aims to synthesize the strengths of these approaches within a single, coherent, reference-free framework that also solves the URSLA shortcut.

## 4.2 REFA Variants: From Distribution Matching to Contrastive Alignment

We develop the REFA framework by first analyzing a reference-free adaptation of InfoNCA and then proposing our primary method, REFA-Dynamic, which employs a more powerful contrastive objective.

**REFA-InfoNCA.** As a baseline for reference-free multi-preference alignment, we first consider a version of InfoNCA without a reference model, which we term **REFA-InfoNCA**. It optimizes a cross-entropy objective to align the model's output distribution ($p_i^{\text{model}} \propto \pi_\theta(y_i|x)$) with a target distribution derived from rewards. While simple, its stationary point occurs when $p^{\text{model}} = p^{\text{target}}$. This can be suboptimal for alignment, as it may require the model to continue assigning non-trivial probability to undesirable responses. A detailed analysis is provided in Appendix F.

**REFA-Dynamic (Our Primary Method).** To create a stronger alignment signal, we introduce **REFA-Dynamic**. This method uses a group-contrastive loss that generalizes the Bradley-Terry model to sets of responses in a reference-free setting. Let the base score for a response be its scaled, length-normalized log-probability, $s(y) = \beta \cdot \overline{\log \pi_\theta}(y|x)$, where $\beta$ is an inverse temperature. The REFA-Dynamic loss is:

$$L_{\text{REFA-Dynamic}}(\theta) = -\log \frac{\sum_{y \in Y^+} e^{s(y)}}{\sum_{y \in Y^+} e^{s(y)} + \gamma \sum_{y \in Y^-} e^{s(y)}}. \tag{6}$$

Here, $\gamma > 0$ adjusts the penalty for the negative set $Y^-$. This formulation directly addresses key requirements for robust alignment:

- **Suppression of Negative Responses:** By placing responses from $Y^-$ only in the denominator, the objective directly incentivizes reducing their assigned probabilities.

- **Collective Uplift of Positive Responses:** All positive responses in $Y^+$ are upweighted in the numerator, reducing the internal competition that arises from purely pairwise or cross-entropy objectives.

Crucially, this contrastive objective possesses a more desirable stationary point. As shown in Appendix I, the loss is minimized as the probabilities of all rejected responses approach zero ($P_\theta(y|x) \to 0$ for all $y \in Y^-$), offering a theoretically superior objective compared to distribution matching.

**W-REFA (Weighted REFA-Dynamic).** For off-policy settings with reliable, fine-grained reward scores, the binary partition can be enhanced. The **W-REFA** variant incorporates deviation-based weighting. The score for each response is additively adjusted: $s_{\text{wtd}}(y) = s(y) + \alpha(r_y - \bar{r})$, where $\alpha$ is a hyperparameter. This variant is particularly effective in our off-policy experiments, while the unweighted REFA-Dynamic shows strong performance in on-policy settings where reward model noise is a greater concern. The full formulation is detailed in the appendix.

## 5 Fine-Grained Length Control via EOS Regularization

The REFA framework developed in the previous section provides a robust, multi-preference objective for reference-free alignment. However, as established in Section 3, the length-normalized scores it relies on are still vulnerable to the URSLA shortcut. In this section, we introduce our primary algorithmic novelty: a family of End-of-Sequence (EOS) probability regularizers designed to counteract this failure mode and provide a versatile mechanism for fine-grained length control.

### 5.1 Counteracting the URSLA Shortcut with Targeted Regularization

The core problem identified is that models can learn to shorten negative responses to satisfy the loss. To prevent this, we introduce a regularizer that penalizes premature termination.

Our primary mechanism, the **Targeted Regularizer**, applies a penalty to the EOS probability of a response, where the penalty's magnitude is proportional to how much shorter the response is than a given target length. In our experiments, we dynamically **set this target to the maximum response length** within each query, which effectively encourages the model

to avoid truncating any response relative to its peers. The regularizer is formulated as:

$$\mathcal{R}_{\text{target}}(\theta) = \sum_{y \in Y^+ \cup Y^-} \lambda P_\theta(\text{EOS at } |y|) \cdot \max(0, \text{TARGET\_LENGTH} - |y|), \tag{7}$$

where $\lambda > 0$ is a hyperparameter scaling the strength of the regularization. By applying this penalty, we disincentivize the model from placing the EOS token early for responses in $Y^-$, directly counteracting the URSLA shortcut, which is especially apparent when we have a *high variance in the response length*, for instance in off-policy responses.

**The Final REFA Loss.** By integrating this regularizer with the REFA-Dynamic objective from Eq. 6, we arrive at the final loss function for our primary method:

$$L_{\text{REFA}}(\theta) = -\log \frac{\sum_{y \in Y^+} e^{s(y)}}{\sum_{y \in Y^+} e^{s(y)} + \gamma \sum_{y \in Y^-} e^{s(y)}} + \mathcal{R}_{\text{target}}(\theta). \tag{8}$$

This complete objective synergistically combines a theoretically sound multi-preference contrast with a targeted mechanism for robust length control, ensuring that alignment is achieved through genuine quality improvements. While this dynamic target $\mathcal{R}_{\text{target}}(\theta)$ is adaptive to peer responses, we note that more adaptive strategies per domain, or per context for setting the target length could be explored in future work. The full training procedure is detailed in Appendix C.

## 5.2 A Set of Tools for Bidirectional and Budgeted Length Control

Beyond solving the URSLA shortcut by encouraging longer responses, the principle of EOS regularization provides a versatile method for controlling output length in either direction to meet specific application needs. We introduce two additional regularizers for such scenarios.

**Budget-Independent Regularizer (for Decreasing Length).** For applications where conciseness is desired (e.g., summarization, tweet generation), the **Budget-Independent Regularizer** can be employed. It is formulated as a penalty on the negative log-probability of the EOS token:

$$\mathcal{R}_{\text{indep}}(\theta) = -\sum_{y \in Y^+ \cup Y^-} \lambda \log P_\theta(\text{EOS at position } |y|). \tag{9}$$

By minimizing this term, the model is encouraged to assign a *higher* probability to the EOS token at its naturally occurring position, promoting more confident and timely termination, which typically leads to shorter, more concise responses.

**Budgeted Regularizer (for Strict Budget Adherence).** For scenarios with strict token budgets due to economic costs or platform constraints, the **Budgeted Regularizer** offers the most fine-grained control. It is designed to penalize premature termination before a specified budget $b$ while incentivizing termination at or after the budget. The formulation creates a "push-pull" dynamic around the budget:

$$\mathcal{R}_{\text{budgeted}}(\theta) = \lambda \left( \frac{1}{b} \sum_{t=1}^{b-1} P_\theta(\text{EOS at token } t) - \frac{1}{|y| - b} \sum_{t=b+1}^{|y|} P_\theta(\text{EOS at token } t) \right). \tag{10}$$

The first term penalizes the average EOS probability mass *before* the budget, discouraging early stops. The second term, with its negative sign, effectively rewards EOS probability mass *after* the budget. This allows practitioners to precisely steer the model's output length to conform to a target budget, directly managing the accuracy-efficiency tradeoff. The effectiveness of these regularizers is demonstrated empirically in Appendix A.

## 6 Experiments

### 6.1 Experimental Setup

We benchmark **REFA** on Mistral-7B and Llama-3-8B across three distinct training paradigms: offline, on-policy, and iterative. Appendix provides a full description of our baselines and experimental configurations.

| Method | Mistral-Base (7B) | | | | Llama-3-Base (8B) | | | |
| --- | --- | --- | --- | --- | --- | --- | --- | --- |
| | AlpacaEval 2 | | Arena-Hard | MT-Bench | AlpacaEval 2 | | Arena-Hard | MT-Bench |
| | LC (%) | WR (%) | WR (%) | GPT-4 | LC (%) | WR (%) | WR (%) | GPT-4 |
| SFT[1] | 8.4 | 6.2 | 1.3 | 6.3 | 6.2 | 4.6 | 3.3 | 6.6 |
| RRHF[1] | 11.6 | 10.2 | 5.8 | 6.7 | 12.1 | 10.1 | 6.3 | 7.0 |
| SLiC-HF[1] | 10.9 | 8.9 | 7.3 | **7.4** | 12.3 | 13.7 | 6.0 | 7.6 |
| DPO[1] | 15.1 | 12.5 | 10.4 | 7.3 | 18.2 | 15.5 | 15.9 | 7.7 |
| IPO[1] | 10.8 | 9.0 | 7.5 | 7.0 | 14.4 | 14.2 | 17.8 | 7.4 |
| CPO[1] | 9.8 | 8.9 | 6.7 | 6.9 | 10.8 | 8.1 | 5.8 | 7.4 |
| KTO[1] | 13.1 | 9.1 | 5.6 | 6.8 | 14.2 | 12.4 | 12.5 | **7.8** |
| ORPO[1] | 14.7 | 12.2 | 7.0 | 7.0 | 12.2 | 10.6 | 10.8 | 7.6 |
| R-DPO[1] | 17.4 | 12.8 | 10.5 | 7.0 | 17.6 | 14.4 | 17.2 | 7.5 |
| MPO[1] | 20.3 | 14.9 | 12.8 | 7.3 | 20.1 | 15.6 | 18.5 | **7.8** |
| SimPO[1] | 21.5 | 20.8 | 16.6 | 7.3 | 22.0 | 20.3 | 23.4 | 7.7 |
| REFA-InfoNCA | 20.4 | 19.1 | 14.2 | 7.1 | 22.8 | 17.1 | 20.2 | 7.5 |
| REFA-1-vs-k | 20.4 | 19.1 | 14.6 | 7.1 | 25.5 | 21.0 | **26.5** | 7.5 |
| REFA-dynamic | 22.8 | 20.4 | 15.7 | 7.3 | 25.6 | 23.1 | 25.4 | **7.8** |
| W-REFA | **24.6** | **23.0** | **16.6** | **7.4** | **26.6** | **24.2** | 23.7 | **7.8** |

Table 1: Comparison of various preference optimization under off-policy setting for baselines on AlpacaEval, Arena-Hard and MT-Bench benchmarks. LC-WR represents length-controlled win rate, and WR represents raw win rate. Best results are in **bold**, second-best are underlined. REFA achieves SOTA performance across all metrics

**Offline Training.** we follow the pipeline established by Zephyr (Tunstall et al., 2023). We first create SFT models by training the base models on UltraChat-200k (Cui et al., 2023). These SFT checkpoints are then used to initialize **REFA** for preference optimization on the static UltraFeedback dataset (Cui et al., 2023).

**On-policy and Iterative Training.** Our on-policy experiments begin with instruction-tuned models Llama-3-8B-Instruct and Mistral-7B-Instruct. To generate preference data aligned with the policy, we synthesize responses for each UltraFeedback prompt (5 candidates, temp=1.0), similar to Meng et al. (2024). These candidates are scored by the Skywork-Reward-Llama-3.1-8B model Liu et al. (2024a), from which we select the top-2 and bottom-2 responses to form the preferred ($Y^+$) and rejected ($Y^-$) sets for **REFA**.

This on-policy data generation is also integrated into iterative framework inspired by SPPO (Wu et al., 2024). The UltraFeedback prompt set is partitioned, and at each round, the current model checkpoint generates fresh responses to be scored and selected. This newly synthesized data is then used to update the model with **REFA** before the next iteration. More details are provided in Appendix K

## 6.2 Experimental Results

| Method | Mistral-Instruct (7B) | | | | Llama-3-Instruct (8B) | | | |
| --- | --- | --- | --- | --- | --- | --- | --- | --- |
| | LC (%) | WR (%) | Arena-Hard | MT-Bench | LC (%) | WR (%) | Arena-Hard | MT-Bench |
| SPPO[1] (Iter 1) | 24.79 | 23.51 | 18.7 | 7.21 | 31.73 | 31.74 | 34.8 | 8.1 |
| REFA (Iter 1) | **27.58** | **28.79** | **25.3** | **7.56** | **37.42** | **38.72** | **34.10** | **7.85** |
| SPPO[1] (Iter 2) | 26.89 | 27.62 | 20.4 | 7.49 | 35.15 | 35.98 | 38.0 | **8.2** |
| REFA (Iter 2) | **31.57** | **36.85** | **26.8** | **7.7** | **47.88** | **50.83** | **42.20** | 8.12 |
| SPPO[1] (Iter 3) | 28.53 | 31.02 | 23.3 | 7.59 | 38.78 | 39.85 | 40.1 | **8.2** |
| REFA (Iter 3) | **34.87** | **40.85** | **23.8** | **7.62** | **52.17** | **60.29** | **52.70** | 8.16 |

Table 2: Results on Mistral-Instruct and Llama-Instruct models in the iterative on-policy setting with Ultrafeedback prompts.

---

[1]These are taken directly from the paper SIMPO(Meng et al., 2024), MPO (Gupta et al., 2025), SPPO (Wu et al., 2024)

**REFA-dynamic outperforms existing preference baselines:** We conducted an extensive comparison of the REFA-Dynamic framework against various preference optimization methods using the Mistral-7B and Llama3-8b for offline setting , Mistral-7B-Instruct and Llama3-8b-Instruct for online and iterative settings, as summarized in Table 1. The results demonstrate that REFA-Dynamic consistently outperforms all baselines across the primary evaluation metrics. **Key Insights:** This highlight the efficacy of REFA-Dynamic in enhancing response quality, reinforcing its state-of-the-art performance in preference optimization.

| Method | Mistral-Instruct (7B) | | | | Llama-Instruct (8B) | | | |
|---|---|---|---|---|---|---|---|---|
| | AlpacaEval 2 | | Arena-Hard | MT-Bench | AlpacaEval 2 | | Arena-Hard | MT-Bench |
| | LC (%) | WR (%) | WR (%) | GPT-4 | LC (%) | WR (%) | WR (%) | GPT-4 |
| SFT | 17.1 | 14.7 | 12.6 | 7.5 | 26.0 | 25.3 | 22.3 | 8.1 |
| DPO[1] | 26.8 | 24.9 | 16.3 | 7.6 | 40.3 | 37.9 | 32.6 | 8.0 |
| IPO[1] | 20.3 | 20.3 | 16.2 | **7.8** | 35.6 | 35.6 | 30.5 | **8.3** |
| KTO[1] | 24.5 | 23.6 | 17.9 | 7.7 | 33.1 | 31.8 | 26.4 | 8.2 |
| ORPO[1] | 24.5 | 24.9 | 20.8 | 7.7 | 28.5 | 27.4 | 25.8 | 8.0 |
| R-DPO[1] | 27.3 | 24.5 | 16.1 | 7.5 | 41.1 | 37.8 | 33.1 | 8.0 |
| SimPO[1] | 32.1 | 34.8 | 21.0 | 7.6 | 44.7 | 40.5 | 33.8 | 8.0 |
| REFA-InfoNCA | 30.2 | 32.1 | 22.7 | 7.6 | 42.8 | 43.5 | 38.9 | 8.0 |
| REFA-1vsk | **33.1** | **36.3** | **26.5** | 7.7 | 49.6 | 50.2 | 43.2 | 8.1 |
| REFA-Dynamic | 32.5 | 34.6 | 25.9 | 7.6 | **49.7** | **50.7** | **44.7** | 8.1 |

Table 3: Comparison of different alignment methods in an *on-policy setting* across multiple evaluation benchmarks on Mistral-Instruct and Llama-Instruct models.

**Effect of EOS Probability Regularization on REFA-dynamic:** We analyzed the effect of the EOS probability regularization term $\lambda$ on the performance of REFA-dynamic on AlpacaEval2 for Mistral-7B-Base in offline-setting with different values of $\lambda$, as shown in Fig. 2. The results reveal that incorporating $\lambda$ significantly improves model performance. As $\lambda$ increases, we observe a notable rise in the Average Response Length, suggesting that higher regularization encourages longer responses.

**Key findings:** Importance of careful tuning of $\lambda$ to balance verbosity and response quality, with moderate values (e.g., $\lambda = 5e-5$) yielding the best overall trade-off across all metrics. Our theoretical analysis in (Appendix J.4) supports this by demonstrating that regularizing the EOS probability stabilizes sequence generation, reducing overly brief responses.

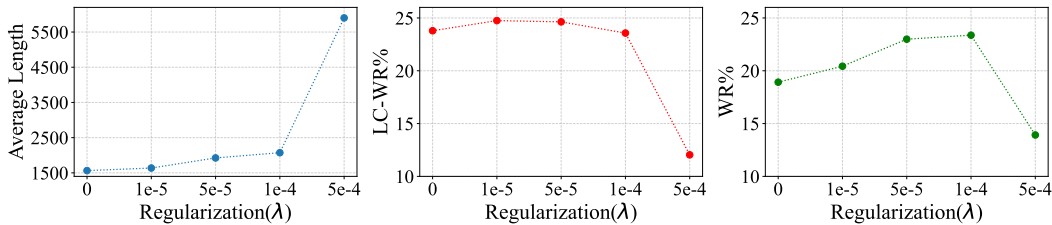

Figure 2: Impact of $\lambda$ on Mistral-Base (7B) Performance: a) Average Response Length, b) Length-Controlled Win Rate (LC-WR), and c) Win Rate (WR).

**Effect of Regularization on Token Efficiency in REFA** Figure 3 shows REFA's effectiveness in controlling response length. Subfigure (a) indicates REFA yields a more balanced length distribution compared to the potential verbosity of InfoNCA or terseness of SimPO. Subfigure (b) confirms that increasing REFA's regularization coefficient $\lambda$ systematically increases average output length, demonstrating $\lambda$ provides fine-grained control over verbosity.

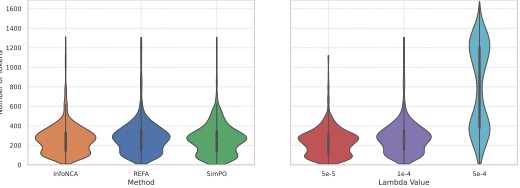

Figure 3: Analysis of token distribution in REFA: (a) across alignment methods (InfoNCA, REFA, and SimPO). (b) across regularization coefficient $\lambda$ for the REFA method.

# 7 Budgeted Length Control

To rigorously evaluate the tradeoff between token efficiency and accuracy, we conducted two key ablation studies focusing on using End-of-Sequence (EOS) token regularization for token reduction. These experiments systematically isolate the impact of different regularization strategies on model performance and output length.

We conducted below experiments on subset of UltraFeedback dataset in online-setting for Llama3-8b-instruct and used AlpacaEval as the downstream evaluation dataset. More extensive experiments are being provided in Appendix *A*

## 7.1 Budget-Independent Regularization

This experiment investigates the impact of budget-independent regularizer $\mathcal{R}_{\text{indep}}(\theta)$ strength, controlled by the hyperparameter $\lambda$. This regularization, $\mathcal{R}_{\text{indep}}(\theta) = -\sum \lambda \log P_\theta(\text{EOS at position } |y|)$, is designed to increase the log-probability of the EOS token. By varying $\lambda$, we evaluated the resulting tradeoff between verbosity and quality. The results demonstrate effective control over response length; as $\lambda$ increases, the average token count decreases, and the entire length distribution shifts towards shorter and precise outputs (Figure 4). Critically, this ablation reveals a non-monotonic relationship between regularization strength and performance.

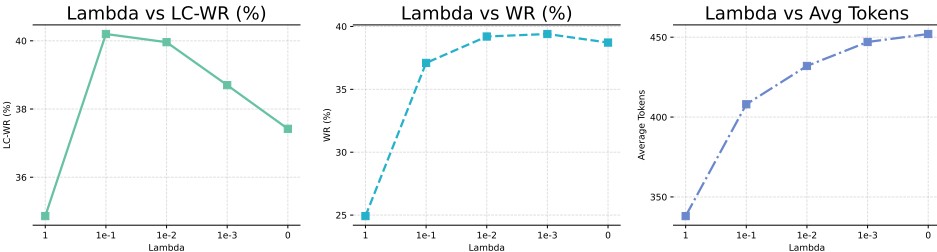

Figure 4: Performance (LC-WR, WR) and Average Tokens vs. $\lambda$ for the budget-independent EOS regularizer on AlpacaEval2. Note the peak LC-WR at $\lambda \approx 0.1$.

## 7.2 Budgeted Regularization

To evaluate the efficacy of precise, targeted length control, we investigates our novel "budgeted" EOS regularizer $\mathcal{R}_{\text{budgeted}}(\theta)$. This experiment demonstrates the regularization strength $\lambda$ across several pre-defined token budgets $b \in \{128, 256, 384, 512\}$. The formulation in eq.10 explicitly penalizes premature termination (EOS before budget $b$) and incentivizes termination at or after it.

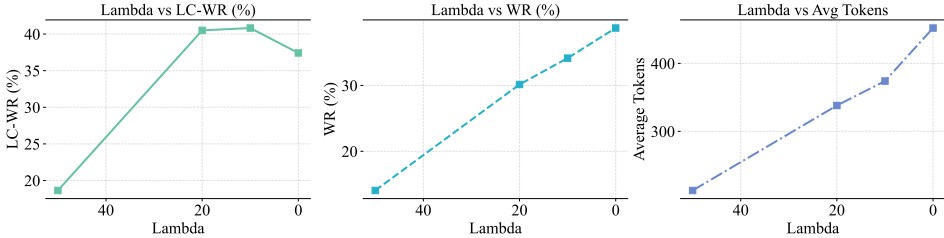

Figure 5: Performance (LC-WR, WR) and Average Tokens vs. $\lambda$ for the budgeted EOS regularizer with a target budget of $b = 256$ on AlpacaEval2.

This approach demonstrates highly precise and effective length control. As $\lambda$ increases, the average token count of generated responses converges directly towards the target budget $b$ shown in figure 5. A consistent accuracy-efficiency tradeoff is observed across all budgets: optimal LC-WR is achieved at a moderate $\lambda$, while overly strict enforcement degrades alignment quality.

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

---

# SUPPLEMENTARY MATERIALS

---

These supplementary materials provide additional details, derivations, and experimental results for our paper. The appendix is organized as follows:

- Section A provides additional experiments for fine-grained length control

- Section B provides a detailed overview of related work pertaining to this paper.

- Section C provides a look on the actual Algorithm used for REFA. This is the primary method studied in Section 6.1

- Section D provides a narrative roadmap to the theoretical results that underpin the REFA framework.

- Section E analyzes the InfoNCA loss function, deriving its gradients, characterizing its stationary points, and examining the role of the reference model.

- Section F provides additional analysis of the InfoNCA loss in a reference-free setting, highlighting issues like promoting low-quality responses and generating contradictory optimization signals.

- Section G details how addressing identified weaknesses in the reference-free InfoNCA formulation leads to a refined, reference-free multi-preference loss function, paving the way for the proposed REFA-base variants.

- Section H introduces the REFA-base loss functions without length normalization, explaining how we incorporate multiple preferences and reward-based weighting to overcome prior limitations.

- Section I focuses on length normalization in REFA, demonstrating how normalizing per-token probabilities prevents trivial short-response biases and encourages genuinely improved responses.

- Section J explores the nuances in the relationship between average per-token uncertainty and sequence length, and justifies the addition of an EOS-probability regularizer to counter subtle length-related incentives.

- Section K More detailed discussion about the Baselines and Evaluation Benchmarks.

- Section L It provides addition ablations done for REFA (Downstream analysis w.r.t $\lambda$, Error-Analysis and statistical-significance, Computational Efficiency)

- Section M It provides qualitative comparison of responses generated from REFA and SIMPO

## A  Additional Fine-Grained Length Control Experiments

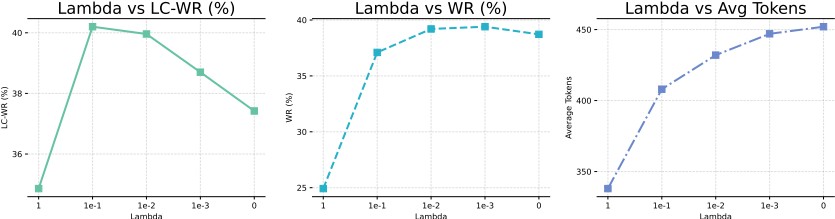

Figure 6: Performance (LC-WR, WR) and Average Tokens vs. $\lambda$ for the budget-independent EOS regularizer on AlpacaEval2. Note the peak LC-WR at $\lambda \approx 0.1$.

## A.1 Impact of Regularization on Performance and Length

Figure 6 demonstrates the relationship between the regularization strength $\lambda$, average response length, and alignment performance (LC-WR and WR). The results show a clear and predictable relationship: as $\lambda$ increases from 0, average length steadily decreases. This controlled increase in length initially leads to better alignment, with performance peaking at a moderate $\lambda \approx 0.1$. For $\lambda > 0.1$, performance declines, suggesting excessive regularization leads to overly verbose and suboptimal outputs.

## A.2 Impact of Regularization on Length Distribution

The effect of the regularizer on the overall distribution of response lengths is visualized in Figure 7. With no regularization ($\lambda = 0$), the model produces a distribution of lengths centered around a lower mean. As $\lambda$ increases, the entire distribution shifts towards shorter responses, with the mean token count increasing accordingly. This visualization confirms that the regularizer does not just affect the average length but systematically influences the model's overall generative behavior to produce longer sequences.

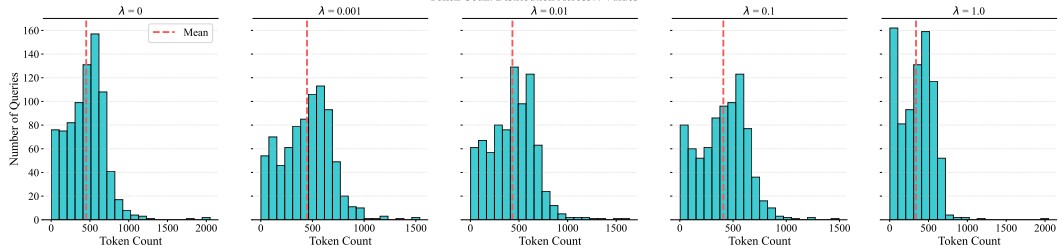

Figure 7: Token count distribution for varying $\lambda$ values using the budget-independent regularizer. As $\lambda$ increases, the mean of the distribution (red dashed line) shifts towards shorter responses.

## A.3 Impact of Budgeted EOS Regularization on Performance and Length

Figures 8, 9, 10, and 11 illustrate the impact of this budgeted EOS regularizer for target length budgets $b = 128$, $b = 256$, and $b = 512$ tokens, respectively. The x-axis represents varying strengths of the regularization parameter $\lambda$. Figures 12, 13, 14, and 15 show the distribution of generated response lengths for different actual $\lambda$ values for $\lambda \in \{0, 10, 20, 50\}$ and target budgets $b \in \{128, 256, 512\}$. The red dashed line indicates the target budget $b$.

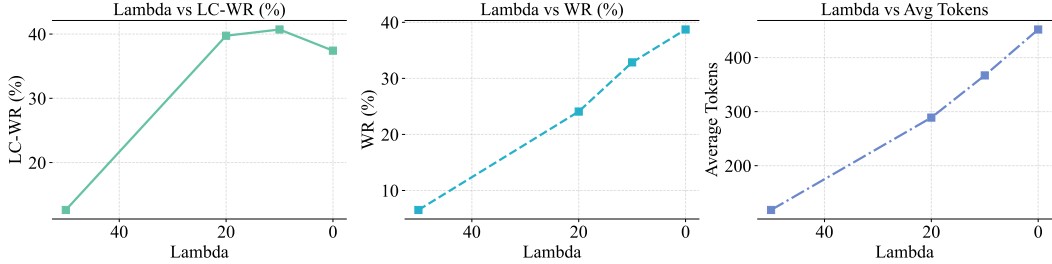

Figure 8: Performance (LC-WR, WR) and Average Tokens vs. $\lambda$ for budget $b = 128$ on AlpacaEval2.

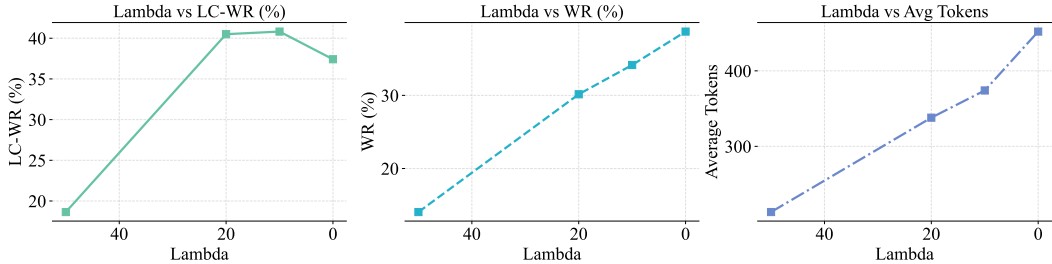

Figure 9: Performance (LC-WR, WR) and Average Tokens vs. $\lambda$ for budget $b = 256$ on AlpacaEval2.

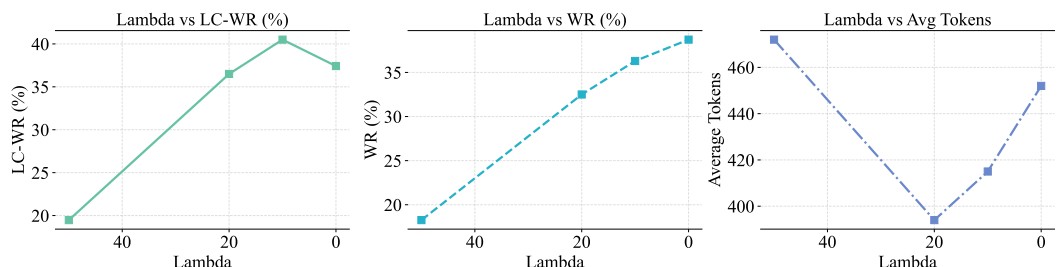

Figure 10: Performance (LC-WR, WR) and Average Tokens vs. $\lambda$ for budget $b = 384$ on AlpacaEval2.

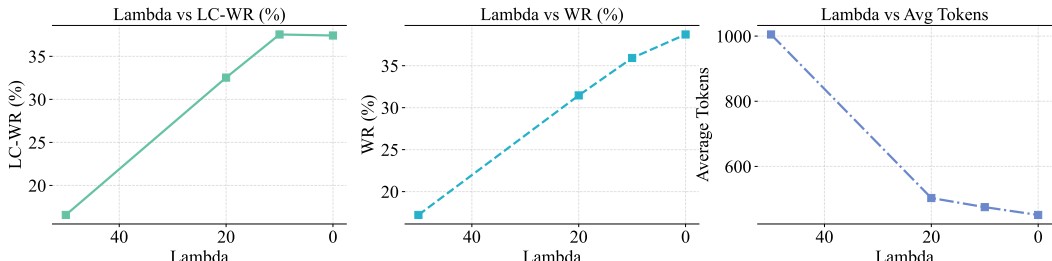

Figure 11: Performance (LC-WR, WR) and Average Tokens vs. $\lambda$ for budget $b = 512$ on AlpacaEval2.

From these figures, we observe a clear **accuracy-efficiency tradeoff**. For each budget $b$, there is an optimal range for $\lambda$ that balances adherence to the budget with achieving high win rates. As $\lambda$ increases, the average token count generally converges towards the target budget, demonstrating effective length control. However, forcing responses to be too strictly confined to a short budget, or allowing excessive verbosity with a large budget, can be suboptimal for alignment quality. This allows practitioners to make informed decisions based on their specific accuracy versus token count requirements.

## A.4   Impact of Budgeted EOS Regularization on Response Length Distribution

Figures 12 -15 show the distribution of generated response lengths for different strengths of $\lambda$ ($\lambda \in \{0, 10, 20, 50\}$) and target budgets $b = 128, 256, 512$. The red dashed line indicates the target budget $b$.

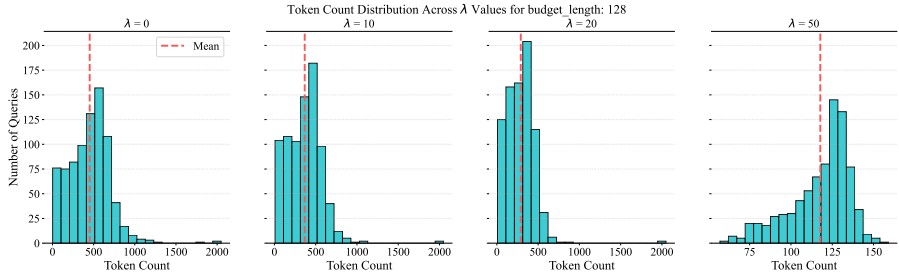

Figure 12: Token count distribution vs. $\lambda$ for budget $b = 128$ on AlpacaEval2.

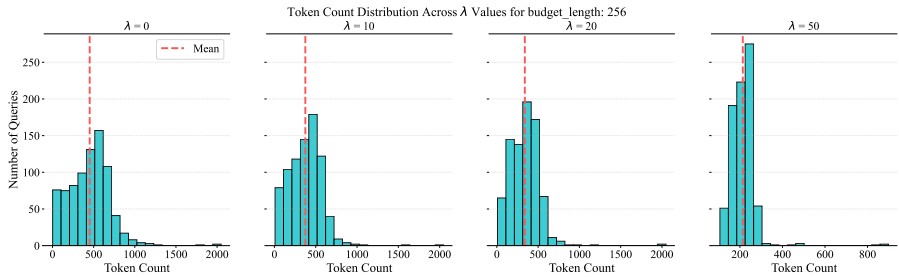

Figure 13: Token count distribution vs. $\lambda$ for budget $b = 256$ on AlpacaEval2.

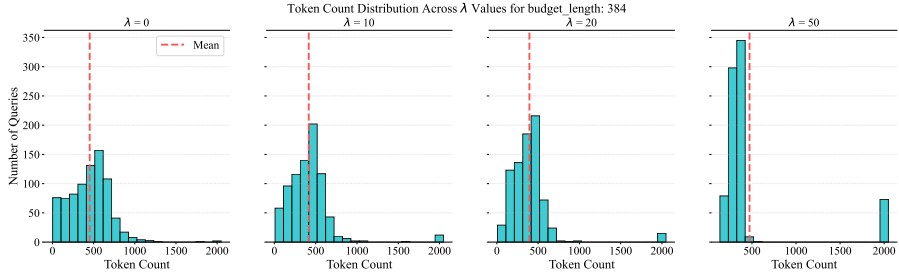

Figure 14: Token count distribution vs. $\lambda$ for budget $b = 384$ on AlpacaEval2.

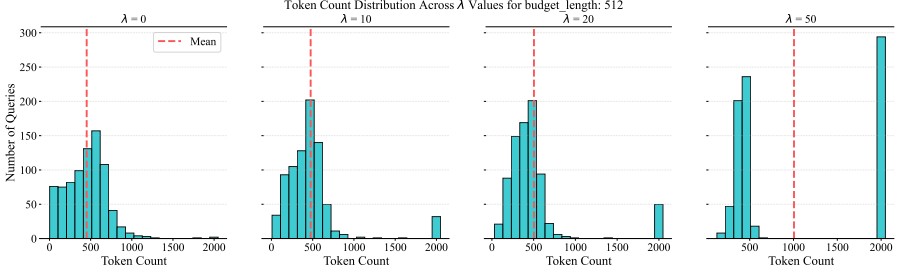

Figure 15: Token count distribution vs. $\lambda$ for budget $b = 512$ on AlpacaEval2.

These distributions clearly show that as $\lambda$ increases from 0 to 50, the response lengths shift and concentrate more tightly around the target budget $b$. This demonstrates the regularizer's effectiveness in penalizing premature termination and pushing generations to be longer and closer to the budget. Yet certain challenges remain.

### A.5 Challenges and Observations with Fixed Budgets on Variable-Length Data

The response length distributions (Figures 12-15, particularly for larger budgets like $b = 384$ and $b = 512$) suggest interesting interactions when applying a fixed budget $b$ to a dataset with inherently high variance in optimal response lengths.

#### A.5.1 Adaptive Learning within Budget Constraints:

The model attempts to adapt its EOS probabilities based on the budget. For instance, if a query naturally warrants a response shorter than a large budget $b$, the regularizer (as formulated in Eq. 10 which penalizes early EOS) might push the model to extend responses, potentially beyond their natural endpoint for that query type if $\lambda$ is high. Conversely, if the query warrants a response much longer than $b$, the model is incentivized to terminate around or after $b$.

#### A.5.2 Potential for Bi-modal or Complex Distributions:

When the inherent optimal lengths for different queries in the dataset vary significantly relative to a single fixed budget $b$, the model's attempt to satisfy the budgeted EOS regularizer across all examples can lead to complex, sometimes bi-modal, length distributions in the aggregate generations. This is because the regularization pressure acts differently depending on whether an unconstrained optimal response would be shorter or longer than $b$.

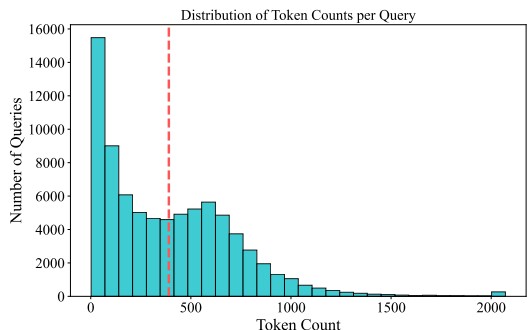

Figure 16: Histogram of Token count vs. Number of Queries on Ultrafeedback.

Figure 16 shows the distribution of our training data, which clearly indicates that there are queries both longer, as well as shorter than the budget. This suggests that while a fixed budget offers control, for datasets with very diverse length requirements, more adaptive budgeting strategies or careful tuning of $\lambda$ might be needed to avoid overly distorting natural response lengths for all query types. Our original, non-budgeted EOS regularizer (Equation 28 in the appendix of the main paper) offers a more dataset-adaptive approach by regularizing based on the EOS position in the training examples themselves.

These findings underscore the power of EOS probability control for managing length but also highlight the nuanced considerations when applying fixed-length targets to diverse datasets. REFA's framework provides the tools for such exploration.

## B Extended Related Work

**Preference Optimization for Alignment.** Early alignment efforts focussed on RLHF through learning an intermediate reward model using reinforcement learning algorithms like PPO (Schulman et al., 2017; Ziegler et al., 2019; Ouyang et al., 2022). While effective, this can be computationally expensive and noisy due to the intermediate reward estimation. Recent approaches, such as Direct Preference Optimization (DPO) (Rafailov et al., 2024b), streamline alignment by directly optimizing a contrastive loss over pairs of preferred and dispreferred responses, bypassing explicit reward models. Subsequent works have extended this idea, exploring variants like IPO (Azar et al., 2023), CPO (Xu et al., 2024a), ORPO (Hong et al., 2024a), and R-DPO (Park et al., 2024), each offering alternative formulations or regularizations. Additionally, methods like RRHF (Yuan et al., 2023), SLiC-HF (Zhao et al., 2023a), GPO (Tang et al., 2024), KTO (Ethayarajh et al., 2024), and SimPO (Meng et al., 2024)

propose diverse preference optimization objectives, some relying on reference models and others operating reference-free.

**Reference-Free Alignment.**    There is growing interest in *reference-free* methods that can directly be used for post-training without a reference model. These approaches avoid the complexity of ratio modeling and can leverage richer data containing multiple candidate responses per query (Cui et al., 2023; Liu et al., 2024b; Zhou et al., 2023; Wang et al., 2024a). Recently, objectives like SimPO (Meng et al., 2024) have shown that focusing on the log-probability of sequences without a separate reference model can achieve strong performance, making the training pipeline simpler and potentially more robust.

**Multi-Preference Optimization.**    The development of multi-preference and reference-free approaches is facilitated by datasets like the UltraFeedback dataset (Cui et al., 2023) that provide scalar rewards corresponding to multiple responses related to a single query. Methods like InfoNCA (Chen et al., 2024a) and MPO (Gupta et al., 2025) use such datasets to generalize beyond pairwise comparisons, simultaneously considering sets of positive and negative responses. Multi-preference objectives better approximate the true preference landscape, better enabling models to estimate the accepted and rejected distributions. While InfoNCA leverages a noise-contrastive framework to align responses according to scalar rewards, MPO introduces deviation-based weighting to emphasize highly positive or highly negative deviations more strongly, thus improving alignment quality.

**Length as a Vector for Reward Hacking.**    A persistent challenge in preference alignment is the tendency for models to engage in "reward hacking" by exploiting spurious correlations in the training data (Gao et al., 2023). Response length is a primary vector for this behavior. It has been widely observed that longer, more verbose responses are often preferred by human and model raters, creating a naive incentive for models to simply "write more" to achieve a higher score (Chen et al., 2024b). Recognizing this, a growing class of preference optimization methods has emerged to explicitly tackle length bias. One prominent approach is *length normalization*, where the implicit reward is based on the average per-token log-probability. This is a core feature in reference-free methods like SimPO (Meng et al., 2024) and ORPO (Hong et al., 2024b), as well as in dedicated works like LN-DPO (Ahrabian et al., 2024) and LMPO (Li et al., 2025). While these methods successfully counter the bias towards verbosity, our work shows they are vulnerable to a second-order exploit, like the URSLA shortcut, highlighting that a more fundamental approach to termination logic is required.

**Efficiency in LLMs: From Model Compression to Inference Budgeting.**    The practical deployment of LLMs is fundamentally constrained by their computational cost. A significant body of research focuses on improving efficiency through **model-centric** approaches. These include reducing model size via quantization (Dettmers et al., 2023), knowledge distillation (Shridhar et al., 2023), and pruning (Bai et al., 2024), as well as developing more efficient architectures like Matryoshka models that allow for adaptive computation (Kusupati et al., 2022; Devvrit et al., 2024). A complementary line of work targets **inference-centric** optimizations like speculative decoding (Leviathan et al., 2023). Our work introduces a third, less explored axis that bridges alignment and efficiency: **output budget control**. By manipulating the generative process at training time to produce outputs of a desired length, we can directly manage the primary drivers of inference cost and latency: the number of generated tokens. REFA contributes to this area by providing a principled mechanism to instill this control during the preference alignment phase itself.

**Positioning REFA.**    REFA builds upon the principles of reference-free, multi-preference optimization established by methods like MPO and SimPO. However, its primary novelty lies in addressing the critical challenge of length-based reward hacking. Unlike prior methods that apply sequence-level normalization or regularization, REFA introduces a novel approach of fine-grained, token-level intervention. By identifying and solving the subtle failure modes introduced by length normalization itself, REFA not only achieves more robust alignment but also provides a practical framework for managing the inference-time budget, directly connecting the goals of preference alignment and computational efficiency.

## C REFA Training Algorithms

This section provides the detailed pseudocode and a step-by-step explanation for the training procedures of the primary REFA variants: the unweighted **REFA-Dynamic** and the weighted **W-REFA**.

### C.1 Algorithm for REFA-Dynamic

Algorithm 1 details the procedure for our primary method. This version uses the un-weighted, group-contrastive loss combined with the Targeted Regularizer. It is designed to be robust and is particularly effective in on-policy and iterative settings where reward model scores may be noisy or less calibrated. The core idea is to use the reward signal to partition responses into positive and negative sets, and then optimize a contrastive objective on the model's length-normalized probabilities, while simultaneously regularizing the model's termination behavior to prevent the URSLA shortcut.

---

**Algorithm 1** REFA-Dynamic Training Procedure

---

**Input:** Dataset $\mathcal{D} = \{(x, \{y_i, r_i\}_{i=1}^K)\}$, learning rate $\eta$, hyperparameters $\beta, \gamma, \lambda$.
**repeat**
    Sample a batch of queries $\mathcal{B} \subset \mathcal{D}$.
    **for all** query $(x, \{y_i, r_i\}_{i=1}^K)$ in the batch $\mathcal{B}$ **do**
        Compute mean reward: $\bar{r} \leftarrow \frac{1}{K} \sum_{i=1}^K r_i$.
        Partition responses: $Y^+ \leftarrow \{y_i \mid r_i > \bar{r}\}, \quad Y^- \leftarrow \{y_i \mid r_i \leq \bar{r}\}$.
        Compute length-normalized log-probabilities $\overline{\log \pi_\theta}(y_i \mid x)$ for all $y_i \in Y^+ \cup Y^-$.
        Compute the base score for each response: $s(y_i) \leftarrow \beta \cdot \overline{\log \pi_\theta}(y_i \mid x)$.
        Compute positive and negative aggregated scores for the contrastive loss:
        $P^+ \leftarrow \sum_{y \in Y^+} e^{s(y)}; \quad P^- \leftarrow \sum_{y \in Y^-} e^{s(y)}$.
        Set target length for regularization: $\text{TARGET\_LENGTH} \leftarrow \max_{y_i \in Y^+ \cup Y^-} |y_i|$.
        Compute the Targeted Regularizer term:
        $\mathcal{R}_{\text{target}}(\theta) \leftarrow \sum_{y \in Y^+ \cup Y^-} \lambda P_\theta(\text{EOS at } |y|) \cdot \max(0, \text{TARGET\_LENGTH} - |y|)$.
        Compute loss for the query: $L_x(\theta) \leftarrow -\log\left(\frac{P^+}{P^+ + \gamma P^-}\right) + \mathcal{R}_{\text{target}}(\theta)$.
    **end for**
    Compute average batch loss: $L_{\text{batch}}(\theta) \leftarrow \frac{1}{|\mathcal{B}|} \sum_{x \in \mathcal{B}} L_x(\theta)$.
    Update parameters: $\theta \leftarrow \theta - \eta \nabla_\theta L_{\text{batch}}(\theta)$.
**until** convergence criteria met

---

### C.2 Explanation of the Procedure.

The REFA-Dynamic training procedure is centered around a composite objective that simultaneously optimizes for preference alignment and robust length control.

First, the algorithm translates raw preference data into a structured signal for a group-wise contrastive loss. For each query, responses are partitioned into positive ($Y^+$) and negative ($Y^-$) sets based on their mean reward. Each response is then scored using its length-normalized log-probability, scaled by an inverse temperature $\beta$ ($s(y) = \beta \cdot \overline{\log \pi_\theta}(y|x)$). This ensures the optimization is sensitive to per-token quality rather than raw sequence length. The core alignment objective, $-\log\left(\frac{P^+}{P^+ + \gamma P^-}\right)$, generalizes the Bradley-Terry model to sets, maximizing the collective probability of the positive set while suppressing the negative set, with the hyperparameter $\gamma$ controlling the weighting strength for the negative examples.

The second component of the objective is the **Targeted Regularizer**, which is critical for counteracting the URSLA shortcut. It applies a penalty, scaled by $\lambda$, to the End-of-Sequence (EOS) probability of any response that is shorter than a dynamically set target (the maximum length in the batch). This mechanism explicitly disincentivizes the model from prematurely

truncating negative responses as a trivial means of satisfying the loss. The final loss is the sum of the contrastive objective and this regularizer. The losses are averaged across a batch before a standard gradient descent step is used to update the model's parameters $\theta$, thereby jointly learning preference alignment and stable length behavior.

### C.3 Algorithm for W-REFA (Weighted REFA-Dynamic)

Algorithm 2 details the procedure for the weighted variant, W-REFA. This version is designed for scenarios, such as off-policy training, where fine-grained and reliable reward scores are available. It enhances REFA-Dynamic by incorporating a deviation-based weighting term directly into the score of each response, allowing the model to prioritize more informative examples (i.e., those with rewards far from the mean).

---

**Algorithm 2** W-REFA (Weighted REFA-Dynamic) Training Procedure

---

**Input:** Dataset $\mathcal{D} = \{(x, \{y_i, r_i\}_{i=1}^{K})\}$, learning rate $\eta$, hyperparameters $\alpha, \beta, \gamma, \lambda$.
**repeat**
  Sample a batch of queries $\mathcal{B} \subset \mathcal{D}$.
  **for all** query $(x, \{y_i, r_i\}_{i=1}^{K})$ in the batch $\mathcal{B}$ **do**
    Compute mean reward: $\bar{r} \leftarrow \frac{1}{K} \sum_{i=1}^{K} r_i$.
    Partition responses: $Y^+ \leftarrow \{y_i \mid r_i > \bar{r}\}, \quad Y^- \leftarrow \{y_i \mid r_i \leq \bar{r}\}$.
    Compute length-normalized log-probabilities $\overline{\log \pi_\theta}(y_i \mid x)$ for all $y_i \in Y^+ \cup Y^-$.
    Compute the weighted score for each response:
    $s_{\text{wtd}}(y_i) \leftarrow \beta \cdot \overline{\log \pi_\theta}(y_i \mid x) + \beta\alpha(r_{y_i} - \bar{r})$.
    Compute positive and negative aggregated weighted scores:
    $P_{\text{wtd}}^+ \leftarrow \sum_{y \in Y^+} e^{s_{\text{wtd}}(y)}; \quad P_{\text{wtd}}^- \leftarrow \sum_{y \in Y^-} e^{s_{\text{wtd}}(y)}$.
    Set target length for regularization: $\text{TARGET\_LENGTH} \leftarrow \max_{y_i \in Y^+ \cup Y^-} |y_i|$.
    Compute the Targeted Regularizer term:
    $\mathcal{R}_{\text{target}}(\theta) \leftarrow \sum_{y \in Y^+ \cup Y^-} \lambda P_\theta(\text{EOS at } |y|) \cdot \max(0, \text{TARGET\_LENGTH} - |y|)$.
    Compute loss for the query: $L_x(\theta) \leftarrow -\log\left(\frac{P_{\text{wtd}}^+}{P_{\text{wtd}}^+ + \gamma P_{\text{wtd}}^-}\right) + \mathcal{R}_{\text{target}}(\theta)$.
  **end for**
  Compute average batch loss: $L_{\text{batch}}(\theta) \leftarrow \frac{1}{|\mathcal{B}|} \sum_{x \in \mathcal{B}} L_x(\theta)$.
  Update parameters: $\theta \leftarrow \theta - \eta \nabla_\theta L_{\text{batch}}(\theta)$.
**until** convergence criteria met

---

**Explanation of Key Differences.** The W-REFA algorithm follows the same overall structure as REFA-Dynamic, with one critical modification in how scores are computed.

- **Line 8 (Weighted Score Computation):** Instead of using only the model's scaled log-probability, the score $s_{\text{wtd}}(y_i)$ is now a composite. It starts with the base score ($\beta \cdot \overline{\log \pi_\theta}(y_i \mid x)$) and then adds a *deviation-based weight term*, $\beta\alpha(r_{y_i} - \bar{r})$.

- **The Role of $\alpha$:** The hyperparameter $\alpha$ scales the influence of the explicit reward signal. A positive $\alpha$ means that responses with rewards far above the mean receive a significant boost to their score, while those far below the mean receive a significant penalty. This creates an implicit curriculum, forcing the model to pay more attention to the most informative, high-deviation examples.

- **Subsequent Steps:** All subsequent steps (Lines 9-12) are analogous to those in Algorithm 1, but they operate on these richer, reward-aware scores ($s_{\text{wtd}}$), thereby propagating the fine-grained reward information throughout the loss calculation.

# D A Reader's Guide to the Theoretical Analysis

This section serves as a narrative roadmap to the formal results that underpin the REFA framework. Our goal is to provide the intuition behind our theoretical claims, explain how they connect, and build a rigorous, step-by-step case for our methodology. The theoretical analysis unfolds in a logical progression: we first deconstruct the properties and pitfalls of a baseline objective (InfoNCA), then use those insights to motivate and analyze our proposed contrastive objective (REFA-Dynamic), and finally, we formalize the subtle length-based shortcut that necessitates our novel EOS regularization, the core algorithmic contribution of this work.

## D.1 The Narrative and Intuition of Our Theoretical Results

**The Flaw in Reference-Based Alignment (Appendix E).** Our theoretical journey begins by justifying the need for a reference-free framework. We analyze the standard, reference-based InfoNCA loss, which aims to match a model's output distribution to a target distribution derived from rewards. Through a formal stationary point analysis, we prove that the presence of the reference model, $\mu(y|x)$, fundamentally alters the optimization target. At equilibrium, the learned policy $\pi_\theta$ does not align directly with the desired preference distribution. Instead, it converges to a skewed distribution proportional to the product of the target probability and the reference model's probability, $p_i^{\text{target}} \mu(y_i|x)$. This result demonstrates that a fixed reference model can act as a permanent, potentially undesirable bias, preventing the policy from ever fully learning the true preference landscape. This provides a strong theoretical motivation for moving to a reference-free setting where the model can learn more directly from the provided preference data.

**Weaknesses of Naive Reference-Free Alignment (Appendix F).** Having established the case for a reference-free approach, we analyze the most direct adaptation: a reference-free version of InfoNCA (which we term REFA-InfoNCA). While its stationary point is an intuitive 'model_distribution = target_distribution', our term-by-term gradient analysis (Lemmas 1 and 2) uncovers critical flaws in its optimization dynamics. We prove that the cross-entropy objective provides a positive gradient push to *all* responses with a non-zero target probability, regardless of their quality. This means the model is inadvertently encouraged to waste capacity learning to generate low-quality responses. Furthermore, the objective creates "gradient conflicts" among high-quality responses, as each is incentivized to increase its own probability at the expense of all others, hindering their collective improvement. This analysis reveals that a simple distribution-matching objective is ill-suited for the noisy, multi-faceted nature of preference alignment.

**The Principled Construction of a Contrastive Objective (Appendix G).** The identified weaknesses of a simple cross-entropy loss motivate the need for a more robust objective. This section of our theory details the systematic, step-by-step process of constructing the REFA-Dynamic loss by addressing each of the identified flaws. We show how moving from a sum of individual objectives to a single, group-contrastive term resolves the issue of gradient conflicts among positive responses. We then demonstrate how partitioning responses into positive and negative sets, and treating them asymmetrically in the loss function (i.e., positives in the numerator, negatives only in the denominator), prevents the model from directly promoting low-quality responses. This structured development shows that the final REFA-Dynamic loss is not an arbitrary formulation but is systematically engineered for robust multi-preference alignment.

**The Theoretical Power of REFA-Dynamic (Appendix I).** With the REFA-Dynamic loss constructed, we then formally prove its theoretical superiority for the task of alignment. Through a detailed gradient analysis of a simplified version of the loss (Lemma 3), we show how the contrastive structure induces different and more desirable dynamics for positive and negative response sets. We formally prove in Lemma 4 that the gradient for any negative response is strictly positive, while the gradient for a positive response is negative. This means the objective has a clear and unambiguous directional influence: it exclusively

pushes to increase the probability of positive responses while simultaneously suppressing the probability of all negative responses. This leads to a far more powerful convergence property: the stationary point of the REFA-Dynamic loss is achieved only as the probability of all negative responses approaches zero ($P_\theta(y|x) \to 0$ for all $y \in Y^-$). This is a much stronger guarantee of alignment than merely matching a target distribution.

**The Necessity of Length Normalization (Appendix I, continued).** Before we can apply the REFA-Dynamic loss, we must first address the foundational issue of scoring. In the same appendix, we prove in Lemma 5 that any objective based on raw, un-normalized log-probabilities is fatally flawed. Because shorter sequences accumulate less negative log-probability, they are assigned artificially high scores. This creates a trivial shortcut where the model can satisfy the loss by simply producing degenerate, short responses. This result rigorously establishes that the use of length-normalized scores is not an optional design choice but a fundamental prerequisite for any valid reference-free preference optimization method.

**The Final Insight: The URSLA Shortcut (Appendix J).** Our theoretical analysis concludes by formalizing the central problem that motivates our primary algorithmic contribution. Having established that length normalization is necessary, we then prove that it creates a new, more subtle problem. We introduce the **Uncertainty Reduction with Sequence Length Assertion (URSLA)** (Conjecture 2), a principle stating that longer coherent sequences tend to have lower per-token uncertainty. Under this principle, we formally prove in Lemma 6 that any length-normalized objective that penalizes negative responses (like REFA-Dynamic) will create a new and perverse incentive: the model is encouraged to *shorten* those negative responses to increase their per-token uncertainty and thus satisfy the loss. This result provides the rigorous theoretical foundation for why an additional, explicit length control mechanism, our EOS regularizer, is an **essential component** for achieving robust, reference-free alignment.

# E Analysis of the InfoNCA Loss

In this section, we provide a detailed analysis of the InfoNCA loss function introduced in previous works. We begin by establishing our notation and problem setup, then proceed to define the InfoNCA loss. Subsequently, we derive its gradients, characterize its stationary points, and examine the implications for the learned policy distribution when a reference distribution is present. Finally, we discuss how removing the reference model recovers a simpler scenario that aligns the learned distribution directly with the target distribution.

## E.1 Notation and Setup

We consider a query (or context) $x \in \mathcal{X}$ and a set of $K$ candidate responses $\{y_i\}_{i=1}^K$. Each response $y_i$ is associated with a scalar reward $r_i = r(x, y_i)$, representing how suitable or aligned the response is according to some evaluation metric or annotated feedback.

We assume access to:

- A policy model $\pi_\theta(y|x)$ parameterized by $\theta$, which assigns a probability to each response $y_i$.
- A reference model $\mu(y|x)$, which provides a baseline distribution over responses. This reference model is fixed and not optimized.

We define the ratio:

$$r_\theta(x, y_i) := \log \frac{\pi_\theta(y_i|x)}{\mu(y_i|x)}. \tag{11}$$

For simplicity, we set $\beta = 1$ in the definitions that follow, but the analysis easily generalizes to arbitrary positive $\beta$.

The target distribution derived from the rewards is given by a softmax transformation:

$$p_i^{\text{target}} := \frac{e^{\alpha r_i}}{\sum_{j=1}^{K} e^{\alpha r_j}}, \tag{12}$$

where $\alpha > 0$ is an inverse temperature-like parameter controlling the sharpness of the target distribution.

## E.2 The InfoNCA Loss

The InfoNCA loss aims to match the distribution induced by $r_\theta(x, y)$ to the target distribution $p_i^{\text{target}}$. Given that $r_\theta(x, y_i) = \log(\pi_\theta(y_i|x)/\mu(y_i|x))$, we define the model distribution as:

$$p_i^{\text{model}} := \frac{e^{r_\theta(x,y_i)}}{\sum_{j=1}^{K} e^{r_\theta(x,y_j)}}. \tag{13}$$

The InfoNCA loss is the cross-entropy between the target distribution $p_i^{\text{target}}$ and the model distribution $p_i^{\text{model}}$:

$$L_{\text{InfoNCA}}(\theta) = -\sum_{i=1}^{K} p_i^{\text{target}} \log p_i^{\text{model}}. \tag{14}$$

Minimizing $L_{\text{InfoNCA}}$ encourages $p_i^{\text{model}}$ to align with $p_i^{\text{target}}$.

## E.3 Gradient Derivation

To understand the optimization dynamics, we compute the gradient of $L_{\text{InfoNCA}}$ with respect to the log-ratios $r_\theta(x, y_i)$.

First, note that:

$$\log p_i^{\text{model}} = r_\theta(x, y_i) - \log\left(\sum_{j=1}^{K} e^{r_\theta(x,y_j)}\right).$$

Differentiating $\log p_k^{\text{model}}$ with respect to $r_\theta(x, y_i)$:

$$\frac{\partial \log p_k^{\text{model}}}{\partial r_\theta(x, y_i)} = \delta_{ik} - p_i^{\text{model}},$$

where $\delta_{ik} = 1$ if $i = k$ and 0 otherwise.

Now, differentiate the loss:

$$\frac{\partial L_{\text{InfoNCA}}}{\partial r_\theta(x, y_i)} = -\sum_{k=1}^{K} p_k^{\text{target}} \frac{\partial \log p_k^{\text{model}}}{\partial r_\theta(x, y_i)}.$$

Substituting the derivative of $\log p_k^{\text{model}}$:

$$\frac{\partial L_{\text{InfoNCA}}}{\partial r_\theta(x, y_i)} = -\sum_{k=1}^{K} p_k^{\text{target}}(\delta_{ik} - p_i^{\text{model}}).$$

Evaluating the sum yields the well-known cross-entropy gradient form:

$$\frac{\partial L_{\text{InfoNCA}}}{\partial r_\theta(x, y_i)} = p_i^{\text{model}} - p_i^{\text{target}}.$$

Thus, the gradient simplifies to the difference between the model and target distributions.

### E.4 Stationary Points

A stationary point occurs where the gradient is zero for all $i$:

$$p_i^{\text{model}} - p_i^{\text{target}} = 0 \quad \implies \quad p_i^{\text{model}} = p_i^{\text{target}} \;\; \forall i.$$

At this stationary point, the ratio distribution $e^{r_\theta(x,y)} / \sum_j e^{r_\theta(x,y_j)}$ perfectly matches the reward-derived target distribution. However, note that $p_i^{\text{model}}$ is defined on the ratio $\pi_\theta(y|x)/\mu(y|x)$, not directly on $\pi_\theta(y|x)$ alone.

**Implications for the Ratio** $\pi_\theta(y|x)/\mu(y|x)$    Recall:

$$p_i^{\text{model}} = \frac{\pi_\theta(y_i|x)/\mu(y_i|x)}{\sum_{j=1}^{K} \pi_\theta(y_j|x)/\mu(y_j|x)}.$$

At the stationary point:

$$p_i^{\text{target}} = p_i^{\text{model}} = \frac{\frac{\pi_\theta(y_i|x)}{\mu(y_i|x)}}{\sum_{j=1}^{K} \frac{\pi_\theta(y_j|x)}{\mu(y_j|x)}}.$$

Rearranging:

$$\frac{\pi_\theta(y_i|x)}{\mu(y_i|x)} = p_i^{\text{target}} \sum_{j=1}^{K} \frac{\pi_\theta(y_j|x)}{\mu(y_j|x)}.$$

Define:

$$A := \sum_{j=1}^{K} \frac{\pi_\theta(y_j|x)}{\mu(y_j|x)}.$$

Then:

$$\pi_\theta(y_i|x) = p_i^{\text{target}} \mu(y_i|x) A.$$

This shows that at equilibrium, $\pi_\theta(y_i|x)$ is proportional to the product $p_i^{\text{target}} \mu(y_i|x)$. Note that we have not assumed $\sum_i \pi_\theta(y_i|x) = 1$; $\pi_\theta(y_i|x)$ could be any nonnegative values. The shape of the solution is defined by:

$$\pi_\theta(y_i|x) \propto p_i^{\text{target}} \mu(y_i|x).$$

If we subsequently impose normalization to interpret $\pi_\theta(y_i|x)$ as a probability distribution, we would set:

$$\hat{\pi}_\theta(y_i|x) = \frac{\pi_\theta(y_i|x)}{\sum_{k=1}^{K} \pi_\theta(y_k|x)} = \frac{p_i^{\text{target}} \mu(y_i|x)}{\sum_{k=1}^{K} p_k^{\text{target}} \mu(y_k|x)}.$$

Define:

$$M := \sum_{k=1}^{K} p_k^{\text{target}} \mu(y_k|x),$$

then:

$$\hat{\pi}_\theta(y_i|x) = \frac{p_i^{\text{target}} \mu(y_i|x)}{M}.$$

In this normalized view, the equilibrium policy distribution aligns with a $\mu$-weighted version of the target distribution rather than $p_i^{\text{target}}$ directly. This may point to a potential improvement over the InfoNCA loss function, simply by removing the reference model, as we show in the subsection below.

### E.5 Removing the Reference Model: A Reference-Free Scenario

The presence of the reference model $\mu(y|x)$ skews the stationary solution. If we were to remove $\mu(y|x)$ from the formulation, i.e., consider a scenario where $r_\theta(x,y) := \log \pi_\theta(y|x)$ without dividing by $\mu(y|x)$, then the model distribution $p_i^{\text{model}}$ would directly be:

$$p_i^{\text{model}} = \frac{\pi_\theta(y_i|x)}{\sum_{j=1}^{K} \pi_\theta(y_j|x)},$$

which is simply the policy distribution itself.

In that reference-free case, setting $p_i^{\text{model}} = p_i^{\text{target}}$ at the stationary point would imply:

$$\pi_\theta(y_i|x) = p_i^{\text{target}},$$

directly aligning the policy distribution with the target distribution. This is the desirable outcome if no baseline or reference distribution is involved. Thus, the introduction of $\mu(y|x)$ in the denominator shifts the stationary solution, necessitating careful design if we want direct alignment with $p_i^{\text{target}}$.

In summary, the InfoNCA loss aligns the ratio $\pi_\theta(y|x)/\mu(y|x)$ to the target $p_i^{\text{target}}$. To achieve direct alignment of $\pi_\theta(y|x)$ with $p_i^{\text{target}}$, a reference-free formulation is required.

## F  Additional Analysis of the InfoNCA Loss Without a Reference Model

In the previous section, we analyzed the InfoNCA loss in the presence of a reference model $\mu(y|x)$. Here, we revisit the InfoNCA formulation under a simpler, reference-free scenario and highlight several issues that arise when optimizing this objective. Specifically, we show how the loss may inadvertently increase the probabilities of low-rated responses and lead to contradictory optimization signals.

### F.1  Reference-Free InfoNCA

Removing the reference model from the formulation amounts to setting:

$$r_\theta(x, y_i) := \log \pi_\theta(y_i|x). \tag{15}$$

In this case, the model distribution becomes:

$$p_i^{\text{model}} = \frac{\pi_\theta(y_i|x)}{\sum_{j=1}^{K} \pi_\theta(y_j|x)}. \tag{16}$$

The InfoNCA loss reduces to the standard cross-entropy between the target and model distributions:

$$L_{\text{InfoNCA}}(\theta) = -\sum_{i=1}^{K} p_i^{\text{target}} \log p_i^{\text{model}}, \tag{17}$$

with $p_i^{\text{target}}$ defined as before.

### F.2  Term-by-Term Gradient Dynamics

**Lemma 1** (Gradient of a Single-Term Objective). Let $\ell_i(\theta)$ be defined as:

$$\ell_i(\theta) := -p_i^{\text{target}} \log p_i^{\text{model}},$$

where

$$p_i^{\text{model}} = \frac{\pi_\theta(y_i|x)}{\sum_{j=1}^{K} \pi_\theta(y_j|x)}$$

and $p_i^{\text{target}}$ is a fixed scalar satisfying $p_i^{\text{target}} > 0$. Assume $\pi_\theta(y_j|x) > 0$ for all $j = 1, \ldots, K$.

Then, the partial derivatives of $\ell_i(\theta)$ with respect to $\pi_\theta(y_i|x)$ and $\pi_\theta(y_j|x)$ for $j \neq i$ are:

$$\frac{\partial \ell_i(\theta)}{\partial \pi_\theta(y_i|x)} = -p_i^{\text{target}} \left( \frac{1}{\pi_\theta(y_i|x)} - \frac{1}{\sum_{j=1}^K \pi_\theta(y_j|x)} \right),$$

and for each $j \neq i$:

$$\frac{\partial \ell_i(\theta)}{\partial \pi_\theta(y_j|x)} = p_i^{\text{target}} \frac{1}{\sum_{k=1}^K \pi_\theta(y_k|x)}.$$

*Proof.* We start from the definition:

$$\ell_i(\theta) = -p_i^{\text{target}} \log p_i^{\text{model}}.$$

Substitute $p_i^{\text{model}}$:

$$p_i^{\text{model}} = \frac{\pi_\theta(y_i|x)}{\sum_{j=1}^K \pi_\theta(y_j|x)}.$$

Thus:

$$\ell_i(\theta) = -p_i^{\text{target}} \left[ \log \pi_\theta(y_i|x) - \log \left( \sum_{j=1}^K \pi_\theta(y_j|x) \right) \right].$$

First, consider the derivative with respect to $\pi_\theta(y_i|x)$:

$$\frac{\partial \ell_i(\theta)}{\partial \pi_\theta(y_i|x)} = -p_i^{\text{target}} \left[ \frac{\partial}{\partial \pi_\theta(y_i|x)} \log \pi_\theta(y_i|x) - \frac{\partial}{\partial \pi_\theta(y_i|x)} \log \left( \sum_{j=1}^K \pi_\theta(y_j|x) \right) \right].$$

Compute each derivative separately:

1. For the first term:

$$\frac{\partial}{\partial \pi_\theta(y_i|x)} \log \pi_\theta(y_i|x) = \frac{1}{\pi_\theta(y_i|x)}.$$

2. For the second term, let $S := \sum_{j=1}^K \pi_\theta(y_j|x)$. Then:

$$\frac{\partial}{\partial \pi_\theta(y_i|x)} \log(S) = \frac{1}{S} \frac{\partial S}{\partial \pi_\theta(y_i|x)} = \frac{1}{S}.$$

Substituting back:

$$\frac{\partial \ell_i(\theta)}{\partial \pi_\theta(y_i|x)} = -p_i^{\text{target}} \left( \frac{1}{\pi_\theta(y_i|x)} - \frac{1}{\sum_{j=1}^K \pi_\theta(y_j|x)} \right).$$

Next, consider $j \neq i$:

$$\frac{\partial \ell_i(\theta)}{\partial \pi_\theta(y_j|x)} = -p_i^{\text{target}} \left[ 0 - \frac{\partial}{\partial \pi_\theta(y_j|x)} \log \left( \sum_{k=1}^K \pi_\theta(y_k|x) \right) \right].$$

Since $\log(S)$ with $S = \sum_{k=1}^K \pi_\theta(y_k|x)$ gives:

$$\frac{\partial}{\partial \pi_\theta(y_j|x)} \log(S) = \frac{1}{S}.$$

Therefore:

$$\frac{\partial \ell_i(\theta)}{\partial \pi_\theta(y_j|x)} = p_i^{\text{target}} \frac{1}{\sum_{k=1}^K \pi_\theta(y_k|x)}.$$

This completes the proof. $\qquad\square$

**Lemma 2** (Directional Influence of a Single-Term Objective). Under the same assumptions as Lemma 1, consider the single-term objective $\ell_i(\theta)$ defined above. To reduce $\ell_i(\theta)$, one should:

- **Increase** $\pi_\theta(y_i|x)$**:** The partial derivative $\frac{\partial \ell_i(\theta)}{\partial \pi_\theta(y_i|x)}$ can be negative if $\pi_\theta(y_i|x)$ is sufficiently small, indicating that increasing $\pi_\theta(y_i|x)$ will decrease $\ell_i(\theta)$.

- **Decrease** $\pi_\theta(y_j|x)$ **for** $j \neq i$**:** For each $j \neq i$, the partial derivative $\frac{\partial \ell_i(\theta)}{\partial \pi_\theta(y_j|x)}$ is strictly positive. Thus, increasing $\pi_\theta(y_j|x)$ raises $\ell_i(\theta)$, while decreasing it lowers $\ell_i(\theta)$.

In summary, minimizing $\ell_i(\theta)$ encourages increasing the probability of the $i$-th response and reducing the probabilities of all other responses.

*Proof.* From Lemma 1, we have:

$$\frac{\partial \ell_i(\theta)}{\partial \pi_\theta(y_i|x)} = -p_i^{\text{target}} \left( \frac{1}{\pi_\theta(y_i|x)} - \frac{1}{\sum_{j=1}^K \pi_\theta(y_j|x)} \right).$$

Since $p_i^{\text{target}} > 0$, the sign of this derivative depends on the relative sizes of $\pi_\theta(y_i|x)$ and $\sum_j \pi_\theta(y_j|x)$. If $\pi_\theta(y_i|x)$ is small compared to the total sum, then:

$$\frac{1}{\pi_\theta(y_i|x)} > \frac{1}{\sum_j \pi_\theta(y_j|x)},$$

making the entire expression negative. A negative derivative w.r.t. $\pi_\theta(y_i|x)$ implies increasing $\pi_\theta(y_i|x)$ reduces $\ell_i(\theta)$.

On the other hand, for $j \neq i$:

$$\frac{\partial \ell_i(\theta)}{\partial \pi_\theta(y_j|x)} = p_i^{\text{target}} \frac{1}{\sum_{k=1}^K \pi_\theta(y_k|x)} > 0.$$

Since $p_i^{\text{target}} > 0$ and the denominator is positive, this derivative is strictly positive. Thus, increasing $\pi_\theta(y_j|x)$ with $j \neq i$ increases $\ell_i(\theta)$, and decreasing $\pi_\theta(y_j|x)$ reduces $\ell_i(\theta)$.

Combining these observations, to minimize $\ell_i(\theta)$, we must increase $\pi_\theta(y_i|x)$ (when beneficial) and decrease $\pi_\theta(y_j|x)$ for all $j \neq i$. This establishes the stated directional influence on the single-term objective. $\square$

## G  Fixing Weaknesses in InfoNCA Style Loss Leads to Reference Free MPO

In the absence of a reference model, the InfoNCA loss reduces to a standard cross-entropy form that aligns the model distribution with a target distribution derived from rewards. Although seemingly straightforward, this formulation exhibits several critical weaknesses when applied to multiple responses with varying quality. In this section, we highlight these issues step-by-step, referencing Lemma 1 and Lemma 2, which describe how each individual loss term behaves, and propose incremental fixes that ultimately motivate a more refined approach.

### G.1  W1: Encouraging Low-Rated Responses

**Issue:** From Lemma 1 and Lemma 2, each single-term objective $\ell_i(\theta)$ encourages increasing $\pi_\theta(y_i|x)$ regardless of the response's quality. Even if $p_i^{\text{target}}$ is small, as long as it is nonzero, the model is pushed to raise the probability of that response, effectively giving low-rated responses unwarranted attention.

**Implication:** Low-quality responses should not receive reinforcement, yet the InfoNCA-style objective provides a positive gradient push as long as $p_i^{\text{target}} > 0$. This introduces noise into training and can skew the model toward suboptimal responses.

*Fix 1: Removing Terms for Low-Rated Responses*

A natural remedy is to remove terms corresponding to low-reward responses. For instance, define a threshold (e.g., mean reward $\bar{r}$) and restrict the loss to responses above this threshold:

$$\widetilde{L}(\theta) := - \sum_{i \in \mathcal{I}_{\text{pos}}} p_i^{\text{target}} \log p_i^{\text{model}}, \quad \mathcal{I}_{\text{pos}} = \{i \mid r_i > \bar{r}\}.$$

Excluding low-rated responses avoids their direct encouragement. However, this approach introduces another weakness.

### G.2 W2: Contradictory Signals Among Good Responses

**Issue:** After removing low-rated responses, only "good" responses remain. Yet, Lemma 2 still applies: each remaining term $\ell_i(\theta)$ attempts to boost its own response's probability while suppressing all others, including other good ones. Consider responses $\{8, 7, 3, 2\}$ and exclude those below the mean. The terms for 8-rated and 7-rated responses now compete against each other, creating contradictory optimization signals.

**Implication:** We want multiple good responses to coexist and improve together. However, the pairwise competitive nature of these single-term objectives prevents a collective uplift. Instead of reinforcing a set of good responses, the model ends up with internal conflicts, slowing training and potentially leading to suboptimal solutions.

*Fix 2: Single "1 vs All" Term*

One idea is to avoid summing over multiple per-response objectives. Instead, choose one top-rated response and form a "1 vs all" loss:

$$L_{\text{1vsAll}}(\theta) = - \log \frac{\pi_\theta(y_{i*}|x)}{\sum_{j=1}^{K} \pi_\theta(y_j|x)},$$

where $y_{i*}$ is the best response. This removes direct conflicts among multiple positive responses since we focus on only one. But what if there are multiple top-tier responses we wish to jointly learn from?

### G.3 W3: Handling Multiple Equally Good Responses is Not Possible with 1 vs All

**Issue:** If there are multiple equally good responses, say $\{8, 8, 1, 1\}$, the "1 vs all" approach cannot leverage both top responses simultaneously. It only promotes a single chosen response $y_{i*}$ at the expense of all others. If we pick the first "8"-rated response, we lose the opportunity to learn from the second "8"-rated one.

**Implication:** The "1 vs all" fix addresses contradictory signals when multiple good responses are present only by ignoring some of them, which is not ideal. We want a mechanism that can consider all top-tier responses together, promoting them as a group.

*Fix 3: MPO-Style Single Term (All Good in Numerator, All in Denominator)*

We can construct a single-term loss function that includes *all* positively rated responses in the numerator and all responses (both positive and negative) in the denominator. This resembles a MPO-like contrastive form:

$$L_{\text{all-pos}}(\theta) = - \log \frac{\sum_{y \in Y^+} \pi_\theta(y|x)}{\sum_{y \in Y^+ \cup Y^-} \pi_\theta(y|x)},$$

where $Y^+$ are all good responses and $Y^-$ are all non-positive responses. This addresses the coexistence issue, but now every positive and negative is treated equally, ignoring the magnitude of deviations.

### G.4 W4: Lack of Weighting

**Issue:** In the MPO-style single-term approach without weighting, all positive responses are treated identically, as are all negative responses. Consider ratings $\{10, 4, 4, 1\}$: 1 is significantly worse than 4, yet both are just "negative" if below the mean threshold. In other cases, like $\{9, 7, 3, 1\}$, 9 and 7 are weighted equally, without learning more from 9 than 7. Without weighting, we fail to differentiate these nuanced gradations in quality.

**Implication:** We want to reward the most outstanding positive responses more than marginally positive ones. Similarly, the truly poor responses should have a stronger negative influence than those that are just below average. Ignoring this leads to suboptimal gradient signals that do not fully exploit the granularity of the reward information.

### *Fix 4: Deviation-Based Weighting*

Assign weights based on the deviation from the mean reward. Responses with higher (positive) deviation get amplified; those with negative deviation are penalized proportionally. This yields:

$$L_{\text{weighted}}(\theta) = -\log \frac{\sum_{y \in Y^+} w_y \pi_\theta(y|x)}{\sum_{y \in Y^+ \cup Y^-} w_y \pi_\theta(y|x)},$$

where $w_y = e^{\alpha \Delta S_y}$ where $\Delta S_y = (r_y - \bar{r})^p$ for some p-value like $p = \{1, 2\}$, where $\bar{r}$ is the mean reward. This weighting refines the loss by making the gradient more informative and aligned with the underlying quality differences.

### G.5 W5: Fine-Grained Control via Hyperparameters

**Issue:** While deviation-based weighting improves the distribution of gradient signals, we may still need more control. Different tasks might require adjusting how heavily to penalize negatives, how sharply to differentiate based on deviations, or how to set an effective "temperature" of the softmax.

**Implication:** A flexible loss function should allow fine-tuning how negatives compete with positives ($\gamma$ parameter), how sensitive it is to rating differences ($\alpha$ parameter), and how sharply it distinguishes between responses ($\beta$ parameter).

### *Fix 5: Hyperparameterized Weighted Multi-Preference Loss*

By introducing hyperparameters $(\alpha, \beta, \gamma)$, we can regulate:

1. $\alpha$: how strongly the deviation affects weighting.
2. $\beta$: an inverse-temperature parameter controlling the softmax sharpness.
3. $\gamma$: a factor that overweights negative responses to ensure a clear contrast.

Incorporating these parameters, we obtain a final, fully adjustable loss function that addresses all previously identified weaknesses, allowing nuanced and stable optimization.

$$L_{\text{MPO}}(\theta) = -\log \frac{\sum_{y \in Y^+} e^{\beta(\pi_\theta(y|x) + \alpha \Delta S_y)}}{\sum_{y \in Y^+} e^{\beta(\pi_\theta(y|x) + \alpha \Delta S_y)} + \gamma \sum_{y \in Y^-} e^{\beta(\pi_\theta(y|x) + \alpha \Delta S_y)}}$$

In summary, each weakness of the InfoNCA-style loss reveals the necessity of more sophisticated mechanisms: filtering out low-rated responses, avoiding contradictory objectives among top responses, considering multiple good responses together, weighting by deviation

for finer quality distinctions, and introducing hyperparameters for greater flexibility. These insights set the stage for refined multi-preference optimization methods that align more closely with the complexity of real-world preference alignment scenarios.

# H  Proposed REFABASE Loss Functions Without Length Normalization

The analysis in the previous sections uncovers several fundamental weaknesses in an InfoNCA-style loss when used without a reference model. We identified:

1. **Encouragement of low-rated responses (W1):** Every response, including low-quality ones, receives upward pressure as long as $p_i^{\text{target}} > 0$, leading to suboptimal reinforcement.

2. **Conflicts among top-tier responses (W2 & W3):** Even after filtering out low-quality responses, multiple good responses compete against each other. This prevents a collective uplift and complicates the optimization dynamics.

3. **Lack of nuanced weighting (W4):** Treating all positive or negative responses equally ignores the fine-grained quality differences indicated by their rewards, missing opportunities to better leverage the available information.

4. **Insufficient flexibility (W5):** Without hyperparameters, one cannot adjust how strongly to penalize negatives, how sensitively to incorporate deviations, or how sharply to distinguish among responses.

These issues highlight the need for a more refined reference-free multi-preference loss function that:

1. **Prioritizes genuinely high-quality responses** without inadvertently promoting low-quality ones.

2. **Reduces direct competition among good responses** so that multiple top-tier candidates can improve collectively.

3. **Incorporates reward-based weighting** to reflect the spectrum of response quality.

4. **Provides hyperparameter-driven flexibility** for controlling emphasis on negatives, sensitivity to rating differences, and the sharpness of the distribution.

## H.1  REFABASE Loss Variants

We propose a family of REFABASE (Reference-Free Alignment) loss functions that build upon these insights. By integrating deviation-based weighting, selective response sets, and hyperparameters for fine-grained control, we address the aforementioned weaknesses. We present two variants:

**REFABASE-1 vs All:**  This variant selects the single highest-rated response $y_{i*}$ defined as:

$$y_{i*} = \arg\max_i r_i, \quad Y^+ = \{y_{i*}\}, \quad Y^- = \{y_j \mid j \neq i^*\}.$$

We assign weights based on deviation from the mean, $\Delta S_i = r_i - \bar{r}$, using an exponential scheme $w_i = e^{\alpha \Delta S_i}$. Introducing $\beta$ for inverse-temperature and $\gamma$ for negative overweighting, the REFABASE-1 vs All loss is:

$$L_{\text{REFABASE-1vsAll}}(\theta) = -\log \frac{e^{\beta(\log \pi_\theta(y_{i*}|x) + \alpha \Delta S_{i*})}}{e^{\beta(\log \pi_\theta(y_{i*}|x) + \alpha \Delta S_{i*})} + \gamma \sum_{y \in Y^-} e^{\beta(\log \pi_\theta(y|x) + \alpha \Delta S_y)}}. \tag{18}$$

This approach eliminates low-rated responses and avoids direct conflict among multiple good responses by focusing solely on the top candidate. However, it cannot simultaneously exploit multiple equally high-quality responses.

**REFABASE-Dynamic:** To incorporate multiple good responses, we define:

$$Y^+ = \{y_i \mid r_i > \bar{r}\}, \quad Y^- = \{y_j \mid r_j \le \bar{r}\}.$$

All positive responses appear in the numerator, and both positive and negative sets appear in the denominator. Applying deviation-based weights and hyperparameters as before, we obtain a *MPO*-style loss:

$$L_{\text{REFABASE-dynamic}}(\theta) = -\log \frac{\sum_{y \in Y^+} e^{\beta(\log \pi_\theta(y|x) + \alpha \Delta S_y)}}{\sum_{y \in Y^+} e^{\beta(\log \pi_\theta(y|x) + \alpha \Delta S_y)} + \gamma \sum_{y \in Y^-} e^{\beta(\log \pi_\theta(y|x) + \alpha \Delta S_y)}}. \quad (19)$$

This REFABASE-dynamic formulation simultaneously addresses all previously identified weaknesses:

- **W1:** Low-rated responses remain in the denominator but do not receive direct promotion.

- **W2 & W3:** Multiple responses are jointly improved, reducing pairwise conflicts.

- **W4:** Deviation-based weighting differentiates among positive responses and among negative ones.

- **W5:** Hyperparameters $(\alpha, \beta, \gamma)$ enable nuanced control over the optimization process.

In essence, the REFABASE-dynamic loss represents the final convergence of our incremental fixes, providing a robust and flexible solution to the shortcomings of the InfoNCA-style loss in reference-free, multi-preference alignment scenarios.

# I  Length Normalization in REFA

In this section, we analyze the behavior of a simplified REFAloss function without hyperparameters or weighting. We focus on the dynamic variant, which selects positive and negative responses based on their rewards relative to the mean. We then highlight how length can become a shortcut for reducing the loss and present a normalization strategy to address this issue.

## I.1  Simplified REFA-Dynamic Loss

Consider the simplified REFA-dynamic loss:

$$L_{\text{REFA-dynamic}}(\theta) = -\log \frac{\sum_{y \in Y^+} \pi_\theta(y|x)}{\sum_{y \in Y^+ \cup Y^-} \pi_\theta(y|x)}, \quad (20)$$

where

- $Y^+ = \{y \mid r_y > \bar{r}\}$ is the set of responses with rewards above the mean $\bar{r}$.

- $Y^- = \{y \mid r_y \le \bar{r}\}$ is the set of responses with rewards below or equal to the mean.

- $\pi_\theta(y|x) = e^{s_y}$, where $s_y = \log \pi_\theta(y|x)$ is the log-probability of response $y$.

Define:

$$Z^+ := \sum_{y \in Y^+} \pi_\theta(y|x), \quad Z := \sum_{y \in Y^+ \cup Y^-} \pi_\theta(y|x).$$

Thus:

$$L_{\text{REFA-dynamic}}(\theta) = -\log \frac{Z^+}{Z} = \log Z - \log Z^+. \quad (21)$$

## I.2 Gradient Analysis of the Simplified REFA-Dynamic Loss

**Lemma 3** (Gradient of the Simplified REFA-Dynamic Loss). For a response $y$ with log-probability $s_y = \log \pi_\theta(y|x)$, the gradient of $L_{\text{REFA-dynamic}}(\theta)$ w.r.t. $s_y$ is:

$$\frac{\partial L_{\text{REFA-dynamic}}(\theta)}{\partial s_y} = \begin{cases} p_y^{\text{model}} - p_y^{\text{pos}}, & \text{if } y \in Y^+, \\ p_y^{\text{model}}, & \text{if } y \in Y^-, \end{cases}$$

where

$$p_y^{\text{model}} = \frac{\pi_\theta(y|x)}{Z}, \quad \text{and} \quad p_y^{\text{pos}} = \frac{\pi_\theta(y|x)}{Z^+}.$$

*Proof.* First, note:

$$L_{\text{REFA-dynamic}}(\theta) = \log Z - \log Z^+,$$

with

$$Z = \sum_{y \in Y^+} \pi_\theta(y|x) + \sum_{y \in Y^-} \pi_\theta(y|x), \quad Z^+ = \sum_{y \in Y^+} \pi_\theta(y|x).$$

Taking derivatives w.r.t. $s_y$:

$$\frac{\partial L}{\partial s_y} = \frac{\partial \log Z}{\partial s_y} - \frac{\partial \log Z^+}{\partial s_y}.$$

Compute each term:

$$\frac{\partial \log Z}{\partial s_y} = \frac{1}{Z}\frac{\partial Z}{\partial s_y}, \quad \frac{\partial \log Z^+}{\partial s_y} = \frac{1}{Z^+}\frac{\partial Z^+}{\partial s_y}.$$

**If $y \in Y^+$:**

$$\frac{\partial Z^+}{\partial s_y} = \pi_\theta(y|x) = e^{s_y}, \quad \frac{\partial Z}{\partial s_y} = \pi_\theta(y|x).$$

Thus:

$$\frac{\partial L}{\partial s_y} = \frac{\pi_\theta(y|x)}{Z} - \frac{\pi_\theta(y|x)}{Z^+} = p_y^{\text{model}} - p_y^{\text{pos}}.$$

**If $y \in Y^-$:**

$$\frac{\partial Z^+}{\partial s_y} = 0, \quad \frac{\partial Z}{\partial s_y} = \pi_\theta(y|x).$$

Thus:

$$\frac{\partial L}{\partial s_y} = \frac{\pi_\theta(y|x)}{Z} - 0 = p_y^{\text{model}}.$$

This completes the proof. □

## I.3 Implications for Positive and Negative Responses

**Lemma 4** (Increasing Probability of Positive Responses Decreases the Loss). To decrease $L_{\text{REFA-dynamic}}(\theta)$, the model must increase $\pi_\theta(y|x)$ for $y \in Y^+$ and avoid increasing $\pi_\theta(y|x)$ for $y \in Y^-$. In particular, raising the probability of positive responses lowers the loss.

*Proof.* Recall the simplified REFA-dynamic loss:

$$L_{\text{REFA-dynamic}}(\theta) = -\log \frac{Z^+}{Z},$$

where

$$Z^+ = \sum_{y \in Y^+} \pi_\theta(y|x), \quad Z = Z^+ + \sum_{y \in Y^-} \pi_\theta(y|x).$$

Since $Y^+ \subseteq Y^+ \cup Y^-$, it follows that $Z^+ \leq Z$, and strictly $Z^+ < Z$ if there is at least one negative response.

**Case $y \in Y^+$:** Notice that:

$$p_y^{\text{model}} := \frac{\pi_\theta(y|x)}{Z}, \quad p_y^{\text{pos}} := \frac{\pi_\theta(y|x)}{Z^+}.$$

Since $Z^+ < Z$, we have $\frac{1}{Z^+} > \frac{1}{Z}$, implying $p_y^{\text{pos}} > p_y^{\text{model}}$. Therefore:

$$\frac{\partial L}{\partial s_y} = p_y^{\text{model}} - p_y^{\text{pos}} < 0.$$

A negative gradient w.r.t. $s_y$ means that increasing $s_y$ (equivalently, increasing $\pi_\theta(y|x)$) for $y \in Y^+$ decreases the loss $L$. In other words, raising the probability of a positive response lowers the loss.

**Case $y \in Y^-$:** If $y \in Y^-$, it does not appear in $Z^+$. Thus:

$$\frac{\partial Z^+}{\partial s_y} = 0, \quad \frac{\partial Z}{\partial s_y} = \pi_\theta(y|x).$$

Then:

$$\frac{\partial L}{\partial s_y} = \frac{\pi_\theta(y|x)}{Z} = p_y^{\text{model}} > 0.$$

A positive gradient w.r.t. $s_y$ indicates that increasing $s_y$ (or $\pi_\theta(y|x)$) for a negative response $y$ raises the loss $L$. To reduce the loss, we must not increase the probabilities of negative responses.

**Conclusion:** To decrease $L_{\text{REFA-dynamic}}$, we must increase the probabilities of positive responses (where the gradient is negative) and avoid increasing the probabilities of negative responses (where the gradient is positive). Thus, minimizing the loss encourages boosting positive responses' probabilities. □

### I.4 Length as a Shortcut

Without length normalization, $\log \pi_\theta(y|x)$ is the sum of token log-probabilities:

$$\log \pi_\theta(y|x) = \sum_{t \in y} \log P_\theta(t|\text{context}).$$

Shorter responses generally have fewer tokens over which to accumulate negative log-probabilities, often resulting in higher $\pi_\theta(y|x)$. Thus, a trivial way to increase $\pi_\theta(y|x)$ for some $y \in Y^+$ is to make $y$ exceedingly short, increasing its probability without genuinely improving per-token quality.

**Lemma 5** (Length-Induced Probability Inflation). *Given two responses $y_1$ and $y_2$ with similar average token probabilities, if $|y_1| < |y_2|$, then typically $\pi_\theta(y_1|x) > \pi_\theta(y_2|x)$, incentivizing shorter responses as a shortcut to reduce $L_{\text{REFA-dynamic}}(\theta)$.*

*Proof.* If each token in both responses has an expected log-probability around $-c$ for some $c > 0$, then:

$$\log \pi_\theta(y_1|x) \approx -|y_1|c, \quad \log \pi_\theta(y_2|x) \approx -|y_2|c.$$

If $|y_1| < |y_2|$, then $-|y_1|c > -|y_2|c$ and thus:

$$\pi_\theta(y_1|x) = e^{-|y_1|c} > e^{-|y_2|c} = \pi_\theta(y_2|x).$$

This shows shorter responses achieve higher probabilities, providing an artificial advantage. □

### I.5   Fix: Length Normalization

To prevent the model from exploiting this length-based shortcut, we introduce length normalization. Instead of:

$$\log \pi_\theta(y|x) = \sum_{t \in y} \log P_\theta(t|\text{context}),$$

we define:

$$\overline{\log \pi_\theta}(y|x) = \frac{1}{|y|} \sum_{t \in y} \log P_\theta(t|\text{context}),$$

and use $e^{\overline{\log \pi_\theta}(y|x)}$ in place of $\pi_\theta(y|x)$ in the loss. This ensures that shorter responses do not gain an undue advantage from having fewer tokens, compelling the model to improve the quality per token rather than simply reducing response length.

### I.6   Modified Loss Function

Given the above length normalization fix, we can now modify our loss function to use the new length normalized log-perplexity values, rather than simply the log-perplexities. We write this as follows:

$$L_{\text{REFA-dynamic}}(\theta) = -\log \frac{\sum_{y \in Y^+} e^{\beta(\overline{\log \pi_\theta}(y|x) + \alpha \Delta S_y)}}{\sum_{y \in Y^+} e^{\beta(\overline{\log \pi_\theta}(y|x) + \alpha \Delta S_y)} + \gamma \sum_{y \in Y^-} e^{\beta(\overline{\log \pi_\theta}(y|x) + \alpha \Delta S_y)}}. \tag{22}$$

## J   Relationship between Average Uncertainty and Sequence Length

Despite our length normalization, we find that the loss function given by 22 still has problems due to a nuanced observation. We will state this observation here as a conjecture, and verify it through our experiments. But before we state this observation, we provide some notations.

**Notation:** Let $y$ denote a response (sequence of tokens) with length $\text{len}(y)$ and token-level conditional probabilities $P_\theta(t \mid \text{context})$. We define the average per-token negative log-probability (perplexity) of $y$ under the model $\pi_\theta$ as:

$$\overline{LP}(y) := \frac{1}{\text{len}(y)} \sum_{t \in y} \big[ -\log P_\theta(t \mid \text{context}) \big]. \tag{23}$$

This quantity $\overline{LP}(y)$ measures the average uncertainty or difficulty that the model $\pi_\theta$ has in predicting the tokens of $y$. Lower $\overline{LP}(y)$ indicates higher model confidence on a per-token basis.

### J.1   Model Uncertainty Reduces with Sequence Length

**Conjecture 2** (Uncertainty Reduction with Sequence Length Assertion (URSLA))**.** Consider a set of responses $y$ generated by the model $\pi_\theta$. Let $\overline{LP}(y)$ be the average per-token negative log-probability of $y$. The URSLA Conjecture states that for a non-empty subset of responses $\mathcal{Y}$, as the length of a response $y \in \mathcal{Y}$ increases, the expected value of $\overline{LP}(y)$ decreases. Formally, there exists a set $\mathcal{Y}$ such that for all $y \in \mathcal{Y}$:

$$\frac{\partial \overline{LP}(y)}{\partial \, \text{len}(y)} < 0, \quad \text{for all } y \in \mathcal{Y}. \tag{24}$$

In other words, longer sequences from $\mathcal{Y}$ are generally easier to predict on a per-token level, yielding lower average negative log-probability than shorter sequences.

This conjecture reflects the empirical observation that models, once committed to a certain semantic or syntactic trajectory in generating text, tend to predict subsequent tokens with increasing ease. We have experimentally verified this phenomenon for the models under consideration.

We note that this conjecture may fail under certain conditions such as when the model encounters domain shifts, incoherent sequences, or adversarial prompts, where longer sequences do not become inherently easier to predict. Empirical verification and domain-specific analysis are thus required to support or refute this conjecture in practical settings.

### J.2 Loss Reduction Reduces Expected Sequence Length

Let $\mathcal{Y}^-$ be a set of "negative" responses, as identified by some alignment criterion, where the training objective encourages decreasing their probabilities. Minimizing the objective often translates into increasing $\overline{LP}(y)$ for $y \in \mathcal{Y}^-$. Under the URSLA Conjecture 2, shorter sequences have higher $\overline{LP}(y)$, implying that reducing response lengths for these negative instances is a straightforward way to achieve the desired increase in per-token perplexity and decrease in their probabilities.

**Lemma 6** (Length Reduction for Negative Responses). Consider a loss function $L(\theta)$ that depends on the probabilities of a set of negative responses $\mathcal{Y}^-$. Suppose that reducing the probabilities of these negative responses (equivalently, increasing their average negative log-probability $\overline{LP}(y)$) decreases the overall loss $L(\theta)$. Under Conjecture 2, there exists a positive correlation between decreasing $\text{len}(y)$ for $y \in \mathcal{Y}^-$ and reducing $L(\theta)$. Formally, let:

$$\frac{\partial L(\theta)}{\partial \text{len}(y)} > 0 \quad \text{for all } y \in \mathcal{Y}^-. \tag{25}$$

Then, minimizing $L(\theta)$ w.r.t. $\theta$ encourages decreasing $\text{len}(y)$ for $y \in \mathcal{Y}^-$.

**Proof:** Assume that for each $y \in \mathcal{Y}^-$, the training objective encourages a reduction in $\pi_\theta(y|x)$, the probability assigned to $y$. Equivalently, this corresponds to increasing $-\log \pi_\theta(y|x)$, or increasing its average negative log-probability $\overline{LP}(y)$. By the chain rule, we have:

$$\frac{\partial L(\theta)}{\partial \text{len}(y)} = \frac{\partial L(\theta)}{\partial \overline{LP}(y)} \cdot \frac{\partial \overline{LP}(y)}{\partial \text{len}(y)}. \tag{26}$$

Since $L(\theta)$ decreases when $\overline{LP}(y)$ increases (to reduce $\overline{\log \pi_\theta}(y|x)$), we have:

$$\frac{\partial L(\theta)}{\partial \overline{LP}(y)} < 0. \tag{27}$$

By the URSLA Conjecture 2,

$$\frac{\partial \overline{LP}(y)}{\partial \text{len}(y)} < 0. \tag{28}$$

Combining the two inequalities:

$$\frac{\partial L(\theta)}{\partial \text{len}(y)} = \frac{\partial L(\theta)}{\partial \overline{LP}(y)} \cdot \frac{\partial \overline{LP}(y)}{\partial \text{len}(y)} > 0, \tag{29}$$

since the product of two negative numbers is positive.

A positive gradient $\frac{\partial L(\theta)}{\partial \text{len}(y)} > 0$ implies that reducing $\text{len}(y)$ decreases $L(\theta)$. Hence, during training, the model is incentivized to shorten negative responses $y \in \mathcal{Y}^-$. This completes the proof. $\square$

### J.3 Discussion on the Surprising Relationship between Sequence Length and Average Model (Un)Certainty

Lemma 6 demonstrates that under the URSLA Conjecture 2, minimizing a loss function that penalizes high-probability negative responses effectively encourages the model to produce shorter negative responses. While length normalization techniques are often introduced to mitigate trivial solutions (such as producing degenerate short outputs), the above analysis reveals that under certain assumptions about length and certainty, length normalization alone can inadvertently incentivize undesirable length reductions. This necessitates additional considerations, such as length-regularization or more sophisticated constraints, to ensure that the model's improvements are not merely a byproduct of changing response lengths but reflect genuine quality enhancements.

### J.4 Justification for an EOS-Probability Regularizer

Real-world training data often reflect inherent human biases. Humans typically write and record relatively concise content due to time, effort, or domain constraints. However, at inference time, users may request more elaborate, detailed, and longer responses from a model, especially when studying complex topics that require extended reasoning or rich explanations. This discrepancy can create a mismatch between the training distribution and the desired inference-time behavior.

**Dataset-Induced Bias for Shorter Sequences.** Human-generated training data often skews toward brevity. Tasks such as writing quick responses, providing minimal clarifications, or answering direct questions do not always require exhaustive elaboration. Economic factors, annotation fatigue, and cognitive load mean human annotators or content creators produce shorter sequences that still suffice to convey basic meaning. As a result, the training corpus is filled with many shorter samples, unintentionally leading the model to treat shorter responses as "normal" or even optimal.

**Mismatch Between Training and Inference Requirements.** While training data may be efficient and minimal, users in deployment scenarios often seek more extensive answers. For instance, a user querying a large language model about a technical topic may benefit from a thorough explanation rather than a terse summary. Similarly, users might expect models to produce intermediate reasoning steps, citations, code snippets, or detailed examples. When the model is trained primarily on shorter samples, it fails to internalize the incentive to produce these richer, longer responses at inference time.

**Premature EOS as a Shortcut.** In the absence of constraints, a model that has learned from predominantly short examples can exploit early termination as a trivial optimization. By producing an end-of-sequence (EOS) token sooner, it can quickly finalize its response and avoid the complexity of generating lengthy, coherent content. This behavior reduces computational effort and under certain loss formulations may even appear optimal, as shorter sequences are well-represented and lower-risk in the model's learned distribution.

**Encouraging Richer Outputs via EOS Probability Control.** To counteract the dataset-induced bias and prevent premature truncation, we introduce a regularizer that influences the probability distribution over the EOS token. By penalizing or rewarding the model for EOS placement, we can realign training incentives with deployment-time requirements. For example, if human evaluators or usage logs indicate a preference for more elaborate answers, we can discourage too-early EOS predictions, nudging the model toward providing fuller, more informative content. Conversely, if the model tends to ramble or produce unnecessarily long responses, the regularizer can be adjusted to encourage timely conclusions.

In essence, the EOS-probability regularizer acts as a corrective mechanism. It helps balance the model's learned tendency toward short outputs, an artifact of the training data, with the user's practical need for more extensive, detailed responses at inference time. By bridging the gap between the dataset distribution and real-world user preferences, we ensure that the model's output quality and length better reflect the desired usage scenarios.

### J.5 Incorporating the Regularizer into the REFA Loss

We start from the previously defined length-normalized REFA-dynamic loss (with deviation-based weights and hyperparameters $\alpha$, $\beta$, $\gamma$ as described in Section H). Let:

$$\overline{P^+} := \sum_{y \in Y^+} e^{\beta(\overline{\log \pi_\theta}(y|x) + \alpha \Delta S_y)}, \quad \overline{P^-} := \sum_{y \in Y^-} e^{\beta(\overline{\log \pi_\theta}(y|x) + \alpha \Delta S_y)}.$$

Then, the REFA-dynamic loss without regularization is:

$$L_{\text{REFA-dynamic}}(\theta) = -\log \frac{\overline{P^+}}{\overline{P^+} + \gamma \overline{P^-}}. \tag{30}$$

To incorporate a preference on the EOS probability, we define a regularization term $\mathcal{R}(\theta)$ that depends on $P_\theta(\text{EOS} \mid \text{tokens before EOS in } y)$. If the training data skew toward shorter sequences, we may increase $\lambda$ to discourage prematurely high EOS probabilities, thus encouraging the model to produce more extensive responses. Conversely, if the model generates unnecessarily long and uninformative content, we can adjust the regularizer to incentivize more timely endings.

A simple choice for such a regularizer is:

$$\mathcal{R}(\theta) = \sum_{y \in Y^+ \cup Y^-} \lambda P_\theta(\text{EOS at position } |y|), \tag{31}$$

where $\lambda$ is a small constant that scales the influence of this regularizer. A positive $\lambda$ can encourage the model to place appropriate probability mass on the EOS token *at the position it appears in the human-written training examples*, gradually adjusting the model's length preferences.

Integrating this regularizer into the loss function, we obtain:

$$L_{\text{REFA-dynamic-reg}}(\theta) = -\log \frac{\overline{P^+}}{\overline{P^+} + \gamma \overline{P^-}} + \mathcal{R}(\theta). \tag{32}$$

By tuning $\lambda$ and the form of $\mathcal{R}(\theta)$, practitioners can mitigate the dataset-induced length bias, steering the model toward producing responses of a more desirable length and detail level. This balanced approach ensures that the model's performance at inference time more closely aligns with user expectations, enhancing both the quality and relevance of its outputs.

## K   Experiments Details

### K.1   Experimental Setup

**Evaluation Benchmarks**   We evaluate our models using three widely recognized open-ended instruction-following benchmarks: MT-Bench, AlpacaEval 2, AlpacaEval and Arena-Hard v0.1. These benchmarks are commonly used in the community to assess the conversational versatility of models across a diverse range of queries.

AlpacaEval 2 comprises 805 questions sourced from five datasets, while MT-Bench spans eight categories with a total of 80 questions. The recently introduced Arena-Hard builds upon MT-Bench, featuring 500 well-defined technical problem-solving queries designed to test more advanced capabilities.

We adhere to the evaluation protocols specific to each benchmark when reporting results. For AlpacaEval 2, we provide both the raw win rate (WR) and the length-controlled win rate (LC), with the latter being designed to mitigate the influence of model verbosity. For Arena-Hard, we report the win rate (WR) against a baseline model. For MT-Bench, we present the scores as evaluated by GPT-4-Preview-1106, which serve as the judge model.

**Baselines** We compare our approach against several established offline preference optimization methods, summarized in Table . Among these are RRHF Yuan et al. (2023) and SLiC-HF Zhao et al. (2023b), which employ ranking loss techniques. RRHF uses a length-normalized log-likelihood function, akin to the reward function utilized by SimPO Meng et al. (2024), whereas SLiC-HF directly incorporates log-likelihood and includes a supervised fine-tuning (SFT) objective in its training process.

IPO Azar et al. (2023) presents a theoretically grounded approach that avoids the assumption made by DPO, which treats pairwise preferences as interchangeable with pointwise rewards. CPO Guo et al. (2024), on the other hand, uses sequence likelihood as a reward signal and trains jointly with an SFT objective.

ORPO Hong et al. (2024b) introduces a reference-free odds ratio term to directly contrast winning and losing responses using the policy model, also incorporating joint training with the SFT objective. R-DPO Park et al. (2024) extends DPO by adding a regularization term that mitigates the exploitation of response length.

InfoNCA Chen et al. (2024a), which introduces a K-category cross-entropy loss, reframes generative modeling problems as classification tasks by contrasting multiple data points. It computes soft labels using dataset rewards, applying a softmax operation to map reward values into probability distributions.

Lastly, SimPO Meng et al. (2024) leverages the average log probability of a sequence as an implicit reward, removing the need for a reference model. It further enhances performance by introducing a target reward margin to the Bradley-Terry objective, significantly improving the algorithm's effectiveness.

| Model Name | Method | Hyper-parameters | | | | |
|---|---|---|---|---|---|---|
| | | $\alpha$ | $\beta$ | $\lambda$ | $\gamma$ | **LR** |
| Mistral-7b-base | REFA-InfoNCA | 0.01 | 3.0 | 5.0e-5 | 2.0 | 3.0e-7 |
| | REFA-1vsk | 0.01 | 2.0 | 5.0e-5 | 2.0 | 3.0e-7 |
| | REFA-Dynamic | 0.01 | 3.0 | 5.0e-5 | 2.0 | 3.0e-7 |
| Llama-3-8b-base | REFA-InfoNCA | 0.01 | 4.0 | 2.0e-5 | 2.0 | 6.0e-7 |
| | REFA-1vsk | 0.01 | 2.0 | 2.0e-5 | 2.0 | 6.0e-7 |
| | REFA-Dynamic | 0.01 | 4.0 | 2.0e-5 | 2.0 | 6.0e-7 |
| Mistral-7b-instruct | REFA-InfoNCA | 1.0 | 2.5 | 1.0e-5 | 4.0 | 3.0e-7 |
| | REFA-1vsk | 1.0 | 2.5 | 1.0e-5 | 3.0 | 3.0e-7 |
| | REFA-Dynamic | 1.0 | 2.5 | 1.0e-5 | 4.0 | 3.0e-7 |
| Llama-3-8b-instruct | REFA-InfoNCA | 1.0 | 10.0 | 1.0e-5 | 4.0 | 4.0e-7 |
| | REFA-1vsk | 1.0 | 10.0 | 1.0e-5 | 3.0 | 4.0e-7 |
| | REFA-Dynamic | 1.0 | 10.0 | 1.0e-5 | 4.0 | 4.0e-7 |
| Mistral-7b-instruct | REFA-Dynamic (Iter1) | 1.0 | 2.5 | 1.0e-5 | 4.0 | 8.0e-7 |
| | REFA-Dynamic (Iter2) | 1.0 | 2.5 | 1.0e-5 | 4.0 | 5.0e-7 |
| | REFA-Dynamic (Iter3) | 1.0 | 2.5 | 1.0e-5 | 4.0 | 3.0e-7 |
| Llama-3-8b-instruct | REFA-Dynamic (Iter1) | 1.0 | 10.0 | 1.0e-5 | 4.0 | 8.0e-7 |
| | REFA-Dynamic (Iter2) | 1.0 | 10.0 | 1.0e-5 | 4.0 | 5.0e-7 |
| | REFA-Dynamic (Iter3) | 1.0 | 10.0 | 1.0e-5 | 4.0 | 5.0e-7 |

Table 4: Hyper-parameter settings for various models and REFA methods. The table is grouped by base models, instruction-tuned models, and iterative tuning runs.

**Hyperparameter Consistency and Tuning Strategy:** The main hyperparameters in REFAare $\alpha$ (reward deviation sensitivity), $\beta$ (inverse temperature), $\gamma$ (negative set penalty), $\lambda$ (EOS regularization strength), and $p$ (power for deviation term, explored as a variant).

Our tuning, performed on a held-out development split of the UltraFeedback dataset (test_prefs), revealed several *consistencies* (see Table 4 for optimal values):

- $\alpha$ **(Deviation Sensitivity):** Showed consistency by base model type: optimal $\alpha = 0.01$ for Base models and $\alpha = 1.0$ for Instruct models.

- $\beta$ **(Inverse Temperature):** Displayed model-family consistency: $\beta = 2.5$ for Mistral-Instruct variants and $\beta = 10.0$ for Llama-3-Instruct variants.

- $\lambda$ **(EOS Regularization):** Consistently optimal at $1.0 \times 10^{-5}$ for Instruct models across variants and iterations; varied slightly for Base models ($2.0 \times 10^{-5}$ to $5.0 \times 10^{-5}$). As a key component for length control, some tuning in a small range is beneficial (see Fig. 2 in main paper).

- $\gamma$ **(Negative Set Penalty):** Generally stable, with $\gamma = 2.0$ often optimal for Base models and iterative REFA-Dynamic. Single-iteration Instruct models used $\gamma \in [3.0, 4.0]$. This parameter did not require extensive fine-tuning.

- $p$ **(Deviation Power):** Explored as a structural variant ($p \in \{0, 1, 2\}$) rather than a continuously tuned hyperparameter. $p = 2$ was often best for REFA-Dynamic, while $p = 0$ (unweighted deviation) also showed strong performance for REFA-1vsk on Llama-3-8B, indicating REFA's core benefits are not solely dependent on this specific weighting.

## L   Additional Ablations

**Effect of Penalty-Ratio $\gamma$ on REFA-dynamic:** To investigate the impact of the penalty-ratio parameter $\gamma$ on the performance of REFA-DYNAMIC, we evaluate two key metrics: LC-WR (%) and WR (%) across varying values of $\gamma$. As shown in Figure 17, LC-WR increases initially, peaking at $\gamma = 3$, before slightly declining. This trend indicates that a moderate penalty ratio balances positive and negative response contributions effectively, improving alignment consistency.

Conversely, WR% shows a decreasing trend as $\gamma$ increases beyond 3, suggesting that overly penalizing negative responses may compromise reward alignment. **Key findings** underline the importance of selecting an optimal $\gamma$ to maintain a balance between alignment consistency and reward signals.

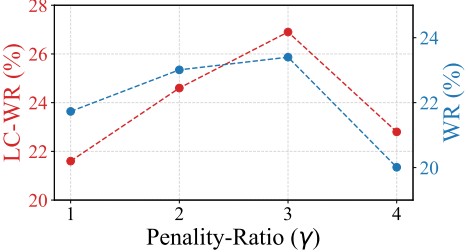

Figure 17: Effect of penalty-ratio $\gamma$ on Mistral-Base (7B) on AlpacaEval2.

**Error Analysis and Statistical Significance** To validate the performance improvements of our REFA framework, we conduct a rigorous error analysis. It is crucial to demonstrate that the observed gains are not artifacts of evaluation variance but represent statistically significant advancements. We analyze the results on two challenging benchmarks, AlpacaEval2 and Arena-Hard, by examining standard errors (SE) and 95% confidence intervals (CI).

The consolidated results in Table 5 provide the foundation for this analysis, comparing our full method and its variants against strong baselines.

**Significance Analysis on AlpacaEval2.**   The standard errors reported for AlpacaEval2 allow us to assess the significance of the performance gap between methods. A common heuristic for statistical significance is a difference between two means that is greater than twice the combined standard error.

- For **Llama-3-8b-Instruct**, the REFA-Dynamic win rate (50.63%) is 12.73 points higher than DPO (37.90%). The difference is substantially larger than their standard errors (1.58 and 1.40), indicating a highly significant improvement. Refa-Dynamic (LC-WR $49.64 \pm 1.58$) also clearly outperforms SimPO (LC-WR $44.70 \pm 1.40$), with a gap of nearly 5 points, which is well over two standard errors of difference.

- For **Mistral-7b-Instruct**, REFA-1vsk (WR 36.23%) achieves a lead of over 11 points against DPO (24.90%). Given the small standard errors (1.48 and 1.30), this margin is again statistically significant.

Table 5: Statistical analysis of Reference-Free Alignment (REFA) and baseline alignment methods on the Llama-3-8b and Mistral-7b instruct models. To assess the reliability of performance gains, we report the length-controlled win rate (LC-WR) and standard win rate (WR) with standard error (SE) on **AlpacaEval2**, and the win rate with its 95% confidence interval (CI) on **Arena-Hard**. The results demonstrate that REFA's improvements are statistically robust and significant.

| Base Model | Method | AlpacaEval2 | | | Arena-Hard |
|---|---|---|---|---|---|
| | | LC-WR (%) $\pm$ SE | WR (%) $\pm$ SE | Win-Rate (%) | WR (%) [95% CI] |
| | DPO | $40.30 \pm 1.40$ | $37.90 \pm 1.40$ | 49.3 | 32.6 [30.3, 34.8] |
| | SimPO | $44.70 \pm 1.40$ | $40.50 \pm 1.40$ | 51.5 | 33.8 [30.9, 35.3] |
| Llama-3-8b-Instruct | REFA-InfoNCA | $43.47 \pm 1.60$ | $44.08 \pm 1.60$ | 53.6 | 38.9 [37.3, 41.4] |
| | REFA-1vsk-Dynamic | $49.56 \pm 1.57$ | $50.23 \pm 1.57$ | 59.8 | 43.8 [42.0, 45.9] |
| | **REFA-Dynamic** | $\mathbf{49.64 \pm 1.58}$ | $\mathbf{50.63 \pm 1.58}$ | **60.2** | **44.7 [42.6, 46.8]** |
| | DPO | $26.80 \pm 1.30$ | $24.90 \pm 1.30$ | 37.1 | 21.0 [18.8, 22.7] |
| | SimPO | $32.10 \pm 1.40$ | $34.80 \pm 1.40$ | 41.3 | 16.3 [15.2, 18.0] |
| Mistral-7b-Instruct | REFA-InfoNCA | $30.20 \pm 1.52$ | $32.10 \pm 1.52$ | 40.2 | 22.5 [21.2, 24.1] |
| | **REFA-1vsk-Dynamic** | $\mathbf{33.12 \pm 1.48}$ | $\mathbf{36.23 \pm 1.48}$ | **43.7** | **26.3 [24.6, 28.1]** |
| | REFA-Dynamic | $32.50 \pm 1.53$ | $34.56 \pm 1.53$ | 42.5 | 25.8 [24.5, 27.6] |

**Significance Analysis on Arena-Hard.** The 95% confidence intervals (CIs) on the Arena-Hard benchmark provide a direct and stringent test of significance.

- For **Llama-3-8b-Instruct**, there is a clear separation between our REFA methods and the baselines. The lower bound of REFA-Dynamic's CI (**42.6%**) is substantially higher than the upper bounds of both DPO (34.8%) and SimPO (35.3%). This non-overlapping interval demonstrates a statistically significant superiority of REFA-Dynamic.

- For **Mistral-7b-Instruct**, REFA-1vsk emerges as the top performer. Its CI of **[24.6%, 28.1%]** does not overlap with DPO's [18.8%, 22.7%] or SimPO's [15.2%, 18.0%]. This confirms that REFA-1vsk achieves a robust and significant win rate advantage on this difficult benchmark.

**Conclusion on Robustness.** The error analysis across both benchmarks and base models confirms the robustness of our findings. The performance gains achieved by the REFA framework, validate REFA as a superior alignment technique.

**Computational Efficiency** We tested the computational efficiency of REFA over a baseline of MPO and show that it achieves significant computational efficiency. Specifically, it leads to a 17% reduction in training time and a 10% decrease in peak GPU memory usage compared to the MPO baseline as show in table 6.

| Metric | REFA | MPO |
|---|---|---|
| Training Time | 4h 1m | 4h 50m |
| Peak GPU Memory | 71 GB | 79 GB |

Table 6: Efficiency comparison of REFA and MPO. The table highlights REFA's advantages in both training time and peak GPU memory consumption.

# M Qualitative Comparison

In this section we provide qualitative comparisons between the baseline model outputs and model outputs from a model trained through the REFA-dynamic approach.

### SimPO vs REFA-Dynamic comparison

**User Query**

*Are you as capable as ChatGPT?*

**SimPO Response**

I am not capable of being as advanced as chatgpt, which is a sophisticated artificial intelligence language model developed by openai. Chatgpt possesses a vast knowledge base, advanced natural language processing capabilities, and continuous learning abilities that enable it to generate human-like responses to various inputs, including text, images, and audio. While I can perform certain language processing tasks, my capabilities are limited compared to chatgpt's advanced neural networks, algorithms, and training data. Nonetheless, I can assist with specific tasks and provide relevant information based on my pre-existing knowledge and programming.

**REFA-Dynamic Response**

I am not capable of being as advanced as chatgpt, which is a sophisticated language model developed by openai. Chatgpt has been trained on vast amounts of textual data, enabling it to generate human-like responses, understand natural language, and provide insights and recommendations based on context and patterns. While I possess natural language processing abilities and can assist with various tasks, I lack the level of sophistication and breadth of knowledge that chatgpt possesses. However, I can continue to learn and improve through training and exposure to new data, making me a useful tool for specific tasks or applications. Ultimately, chatgpt's capabilities are unique and exceptional, and it remains a benchmark for advanced ai language models.

## SimPO vs REFA-Dynamic comparison

### User Query

*If you could help me write an email to my friends inviting them to dinner on Friday, it would be greatly appreciated.*

### SimPO Response

Subject: Let's Gather for a Delicious Dinner on Friday! Dear [FriendsŽ2019 Names], Greetings and warm wishes! I hope this email finds you all doing well and enjoying your week so far. As summer winds down, let's make the most of our time together and plan a delightful dinner gathering on Friday evening. Mark your calendars for Friday, [Date], at [Time], as we invite you to join us for a scrumptious feast at our place. Here are the details: Date: [Date] Time: [Time] Location: [Your Address] We've prepared an exciting menu filled with mouthwatering dishes, including [list some appetizing dishes or cuisine theme]. We can't wait to share our culinary creations with you and indulge in some quality time together. Please RSVP by [Date] so we can ensure adequate food and seating arrangements for everyone. Kindly let us know if you have any dietary restrictions or preferences, and we'll do our best to accommodate them. Looking forward to catching up, sharing laughs, and creating unforgettable memories over a delightful dinner. Bring your favorite drinks, appetizers, or desserts to add to the festive ambiance. Don't miss out on this opportunity to unwind, connect, and savor some delicious food with dear friends. See you soon! Warm regards, [Your Name]

### REFA-Dynamic Response

Subject: Let's Gather for a Delicious Dinner on Friday! Dear Friends, Greetings and warm wishes! I hope this email finds you all doing well and enjoying your week. As summer winds down, let's make the most of it by coming together for a delightful dinner gathering on Friday evening. We're thrilled to invite you to join us for a cozy and intimate dinner at our place on [date] at [time]. We've planned an exciting menu filled with mouthwatering dishes, and we can't wait to share it with your company. Here's a sneak peek: [mention some appetizing highlights or cuisine theme, if applicable]. This casual yet elegant dinner party promises great conversations, laughter, and memorable moments. Please mark your calendars for Friday, [date], and kindly RSVP by [date] so we can prepare accordingly. Feel free to bring your favorite beverage or wine to complement the meal. Let's catch up, indulge in delicious food, and create cherished memories amidst these unprecedented times. We're excited to have you all as our esteemed guests and look forward to seeing you soon. Kindly let us know if you have any dietary restrictions or preferences, and we'll make sure to accommodate them. Please share this invite with anyone who might be interested in joining us. Looking forward to hearing back from you and seeing you all at our dinner table! Warm regards, [Your Name] [Your Contact Information] Let's make this a fantastic evening together! Best regards, [Your Friends' Names] (Optional: Add a fun and personal touch by including a relevant GIF or emoji related to food or gathering, if appropriate.) Note: Adjust the tone, style, and details according to your relationship with your friends and the occasion's specifics. Good luck and enjoy your dinner!

