# OpenReview forum: "REFA: Reference Free Alignment with Fine-Grained Length Control"
_colmweb.org/COLM/2025/Conference — COLM 2025_

### Official Review · Reviewer_zZqs · 2025-04-17

**Rating:** 7
**Confidence:** 5
**Ethics Flag:** 1

**Summary:**

The paper presents a new method for offline alignment which is reference-free, multi-preference and combines 3 different signals: length normalization, deviation-based weighting, and an EOS probability regularizer.

The paper provides a good motivation for each component in its algorithm and a good motivation for each (while justifying the differences to previous work).

I found the writing sometimes a bit "dense" and I can see some whitespacing mistakes that seem like for fitting the paper into 9 pages. E.g: line 290 and 293 there is no space after the full stops.

**Questions To Authors:**

Questions are written above.

**Reasons To Accept:**

The paper seems to show good results on benchmarks that have been used on similar papers (AlpacaEval, MT-Bench, ArenaHard). It also presents a good motivation for each of the components in the loss and how they contribute for the final performance.

**Reasons To Reject:**

I don't have any reason to reject. The authors setup has been standard to compare offline methods such as this. Nonetheless I believe there are improvements to be made to make the setup more robust.

1) What were the  hyper parameters used on each method? Results for these methods are highly sensible to hyper-parameters and the authors acknowledge that but then don't show what they used or what they tried
2) The setup only uses preferences taken from other models (off-policy) but how does their method compare to other methods when data is on-policy? For this I would recommend a different setup where all samples are taken from the SFT model and then ranked according to an open reward model like ArmoRM or Skywork.
3) How does it compare to WPO? WPO is a recent PO method where they actually weight each sample according to the models policy and it seems to achieve also great results in "length-controlled". This is more for my curiosity because I think authors used enough methods.

---

> ### Author Response · Authors · 2025-06-03
> **Response to Reviewer zZqs**
>
> We sincerely thank Reviewer zZqs for their thorough review and positive feedback, particularly on REFA's good results on standard benchmarks and the clear motivation provided for each component of our method. We appreciate this acknowledgment. We have included several suggestions and performed new experiments based on the reviewer's suggestions, and would like to thank the reviewer for these suggestions.
> In fact, **these experiments in the on-policy and iterative settings bring our the strength of the method even more strongly**
>
> We address the points raised for further clarification:
>
> 1.  **Writing Style and Formatting:**
>
> We thank the reviewer for pointing out these formatting issues. We will carefully review and correct all spacing and formatting errors throughout the manuscript for the final version to improve readability (see attached PDF, Rebuttal 1).
>
> 2.  **Hyperparameter Details:**
>
> Our detailed rebuttal (see attached PDF, Rebuttal 2) clarifies our tuning strategy on a development split (UltraFeedback `test_prefs`) and highlights that many REFA hyperparameters show consistent optimal values across different models or settings (e.g., $\alpha$ by model type, $\beta$ by model family, $\lambda$ for Instruct models), reducing the need for exhaustive per-case tuning. A full table of the optimal hyperparameters used for our REFA variants (Table 1 in the attached PDF) is provided, and these details will be added to the paper's appendix.
>
> 3.  **On-Policy Data Comparison:**
>
> This is an excellent suggestion. While our initial experiments focused on the standard offline setting, we have conducted **new experiments in an iterative training paradigm** (see "Additional Experimental Results," Tables 2, 3, & 4 in the attached PDF), which is a common approach to approximate on-policy learning.
>
>
> ### On Policy Setting:
>
>
> These results show REFA variants achieving strong performance in an on-policy style setting (Table 2 in PDF). For instance, REFA-Dynamic (Llama-Instruct) achieves 49.7% LC-WR on AlpacaEval2 and 44.7% WR on Arena-Hard.
>
>
> ### Iterative On-policy Setting:
>
>
> Furthermore, iterative REFA-Dynamic significantly outperforms a strong iterative baseline (SPPO) over 3 iterations (Tables 3 & 4 in PDF), reaching up to **52.17% LC-WR / 60.29% WR on AlpacaEval2** and **52.70% WR on Arena-Hard**. This demonstrates REFA's effectiveness when the training distribution progressively aligns with the evolving model, simulating key aspects of on-policy learning.
>
>
> 4.  **Comparison with WPO:**
>
> We thank the reviewer for the reference to Weighted Preference Optimization (WPO) by \citet{zhou2024wpo}. It is an interesting and relevant recent approach. As detailed in our response (attached PDF, Rebuttal 4), while a direct empirical comparison was not part of our initial design, we will ensure WPO is appropriately cited and discussed in our related work section in the revised manuscript, highlighting its contributions.
>
> ---
>
> We truly thank the reviewer for several positive suggestions relating to the work. Should they be satisfied with our new experiments, we would be grateful if they would consider increasing their score.
>
> [Link to Rebuttal_zZqs.pdf](https://drive.google.com/file/d/19bhHSza2PfSL_DSwN2D3vUEq3ZyHboH9/view)
>
>
> ---
>
> ### Summary of Important New Results:
>
> Our new experiments, detailed in the attached PDF, provide further strong evidence for REFA's capabilities:
>
> **Strong On-Policy/Iterative Performance:**
>
> REFA variants demonstrate significant leads over baselines on AlpacaEval2 and Arena-Hard for both Mistral-Instruct (7B) and Llama-Instruct (8B) in this setting. For instance, Llama-Instruct REFA-Dynamic achieves 49.7% LC-WR (AlpacaEval2) and 44.7% WR (Arena-Hard).
>
> **Significant Gains in Iterative Optimization:**
>
> When applied iteratively, REFA-Dynamic substantially outperforms a strong baseline like SPPO, achieving up to **52.17% LC-WR / 60.29% WR** on AlpacaEval2 and 52.70% WR on Arena-Hard for both Llama-3-8B and Mistral-7B SFT models after three iterations.
>
> **Direct Comparison & Statistical Significance:**
>
> Furthermore, new **direct pairwise comparisons** (detailed in the PDF) show REFA-Dynamic achieving substantial win-rate improvements (e.g., up to 60.6%) over strong baselines like SimPO in on-policy evaluations. Additionally, new **error analysis** (Std. Error and 95% CIs, also in the PDF) confirms the **statistical significance** of REFA's improvements in these on-policy/iterative settings.

---

> > ### Comment · Reviewer_zZqs · 2025-06-06
> >
> > Thanks for including the on-policy experiments and given the strong results I also raised my scores.

---

### Official Review · Reviewer_9iEL · 2025-05-10

**Rating:** 6
**Confidence:** 4
**Ethics Flag:** 1

**Summary:**

The paper proposed an interesting alignment method which can control for various dataset biases.

The experiment setup is good and clean in the paper. However, it makes some wrong assumptions (bias towards brevity) and makes some unnecessary SOTA claims which led to my overall score.

**Reasons To Accept:**

Mathematically interesting approach which is explicitly designed to tackle biases present in preference data.
Good list of related methods to which authors compared their approach.
Good experiment setup: same starting model, same data, vary only approaches.

**Reasons To Reject:**

Two main concerns:
1) The authors start with assumption that existing preference datasets have bias towards brevity. In my experience it is quite the opposite - the longer response is often preferred by humans and, especially by models from GPT-4 family. This has been extensively shown in the literature as well: https://arxiv.org/pdf/2407.01085 (Figure 7) and https://arxiv.org/pdf/2410.16184 (Figure 2).

This is especially true for UltraFeedback dataset which is used in this work. In that data, longer responses are typically preferred.

2) SOTA claims.  Authors claim that they get SOTA. But https://huggingface.co/jieliu/Storm-7B from “Iterative Length-Regularized Direct Preference Optimization: A Case Study on Improving 7B Language Models to GPT-4 Level” is based on Mistral-7B and has 50.5% - 50.3% on AlpacaEval 2 https://tatsu-lab.github.io/alpaca_eval/ (Community) and https://huggingface.co/FuseAI/FuseChat-Gemma-2-9B-Instruct has even higher scores

---

> ### Author Response · Authors · 2025-06-03
> **Addressing Concerns of Reviewer 9iEL**
>
> We sincerely thank Reviewer 9iEL for their thorough review and for recognizing several strengths in our work, including: the proposal of an interesting alignment method for controlling dataset biases, our clean experimental setup, the good list of related methods, and the consistent experimental design (same starting model, data, varying only approaches), and the mathematical strength of our work. We appreciate this positive feedback.
>
> We address the two main concerns raised by the reviewer here:
>
> 1.  **Assumption of Bias Towards Brevity:**
>
> This is in fact a nuanced point. We agree that longer, comprehensive responses are often preferred by both humans and reward models. This has a first order effect on alignment. Our work instead addresses the second order effect relating to length -- i.e. when considering length averaged log-likelihood scores in preference optimization, the *alignment shortcut* to decreasing the probability of rejected responses is to decrease length.
>
> This subtle issue arises from the optimization dynamics (consisting primarily of the following primitives):
>
>     *    Chosen responses are often longer than rejected responses (as pointed out by the reviewer). But longer responses have lower probability.
>     *   Hence, reference free settings (see for example Meng et al., 2024, SimPO) use length normalisation (dividing log-probs by number of tokens) for better optimization.
>     *   However, even with this normalization, our URSLA conjecture (supported by Fig. 1) suggests length of sequence is correlated with higher per-token probabilities, or lower normalised negative log-likelihoods.
>     *   Given DPO strongly penalizes rejected responses, models can learn a shortcut: to make rejected responses appear worse (lower per-token probability) by simply shortening them or terminating them prematurely, thereby satisfying the loss without genuine quality improvement.
>
>     Our EOS regularizer is designed to counteract this specific optimization-induced shortcut, encouraging richer, higher-quality outputs by ensuring improvements stem from content quality rather than length manipulation.
>
>
> **We provide a more detailed step-by-step explanation of this nuanced interplay in the attached PDF (Rebuttal 1)** and kindly request the reviewer to consider this full argument.
>
> 2.  **SOTA Claims:**
>
> We acknowledge the strong models cited. Our original SOTA claims (Table 2 in submission) were for *single-iteration offline reference-free multi-preference alignment* on specific base models/datasets. In our detailed rebuttal (see attached PDF, Rebuttal 2), we clarify this scope and discuss the cited models.
>
>
> **Crucially, we present new iterative results for REFA-Dynamic** (Tables 4-6 in the attached PDF). For Llama-3-8B-Instruct, after 3 iterations, REFA-Dynamic achieves **52.17% LC-WR and 60.29% WR on AlpacaEval2**, and **52.70% WR on Arena-Hard**. These results are highly competitive and in a similar performance tier to the strong iterative methods mentioned (e.g., Storm-7B's ~50.5% WR on AlpacaEval2). This demonstrates REFA's effectiveness when applied iteratively.
>
> We believe these clarifications and new results address the core concerns and significantly strengthen our paper's contributions. We have provided detailed responses and additional experiments in the attached document. Given these clarifications and new supporting data, **we urge Reviewer 9iEL to reconsider their evaluation and would be grateful if they would consider increasing their score.**
>
> Please find attached pdf where we detail our response: [Link to Rebuttal](https://drive.google.com/file/d/1F1IzxnoybEZbAyvCl_Lp6ZZp1qZ-JRwD/view?usp=sharing)
>
> ---
>
> ### Summary of Important New Results:
>
> Our new experiments, detailed in the attached PDF, provide further strong evidence for REFA's capabilities:
>
> **Strong On-Policy/Iterative Performance:**
>
> REFA variants demonstrate significant leads over baselines on AlpacaEval2 and Arena-Hard for both Mistral-Instruct (7B) and Llama-Instruct (8B) in this setting. For instance, Llama-Instruct REFA-Dynamic achieves 49.7% LC-WR (AlpacaEval2) and 44.7% WR (Arena-Hard).
>
> **Significant Gains in Iterative Optimization:**
>
> When applied iteratively, REFA-Dynamic substantially outperforms a strong baseline like SPPO, achieving up to **52.17% LC-WR / 60.29% WR** on AlpacaEval2 and 52.70% WR on Arena-Hard for both Llama-3-8B and Mistral-7B SFT models after three iterations.
>
> **Direct Comparison & Statistical Significance:**
>
> Furthermore, new **direct pairwise comparisons** (detailed in the PDF) show REFA-Dynamic achieving substantial win-rate improvements (e.g., up to 60.6%) over strong baselines like SimPO in on-policy evaluations. Additionally, new **error analysis** (Std. Error and 95% CIs, also in the PDF) confirms the **statistical significance** of REFA's improvements in these on-policy/iterative settings.

---

> > ### Comment · Reviewer_9iEL · 2025-06-08
> >
> > thank you for your response and changes. I've increased my score

---

> ### Comment · Area_Chair_XSEr · 2025-06-07
>
> Dear reviewer,
>
> thank you for your valuable effort. I noticed that you have not answered the authors rebuttal. Could please let us know what are your thoughts and if the answer has been satisfactory? Do you have additional questions?  Thank you!

---

### Official Review · Reviewer_wVBW · 2025-05-13

**Rating:** 6
**Confidence:** 3
**Ethics Flag:** 1

**Summary:**

This paper introduces a reference-free multi-preference optimization method. It combines some of more successful techniques in the literature such as deviation-based weighting and length-normalized scores. In addition, an additional EOS regularization term is introduced to further control the termination of model predictions. On both Mistral and Llama models, the experiments show that the proposed method (REFA-dynamic) significantly outperforms various baselines.

**Questions To Authors:**

1. The main results in Table 2 look the best for AlpacaEval 2, but less so for Arena-Hard, and much less for MT-Bench. I wonder if there's an alternative way to evaluate two systems head-to-head (i.g. pairwise comparison) so that we can conclude method X is significantly better than method Y.
2. Out of curiosity, do you have a plausible explanation for why the trends for Llama-3 looks different than those for Mistral? For instance, REFA-1-vs-k (p=0) performs the best for Llama-3 on Arena-Hard, but not so much for Mistral.
3. The paper is very heavy on analytical and quantitative results, but I think it would benefit greatly from some additional examples to show how the proposed methods might help (e.g., how did EOS regularization help on a specific example?)

**Reasons To Accept:**

1. The proposed method is well-motivated and for the most part straightforward to follow. I find the section on URSLA illuminating and intuitive.
2. The gains appear solid, although see the question section below for one of my concerns.
3. The ablations are good additions, answering many questions I had in mind while reading through the experiment section.

**Reasons To Reject:**

My main reason for rejection concerns the lack of details on the many variants of the proposed methods, all presented in the bottom block of rows in Table 2. It's critical information and I couldn't find relevant information despite multiple readthroughs. For instance, what is "dyamic", "1-vs-k", and how do you combine REFA with "InfoNCA"? For this reason, I'm giving a below-acceptance assessment, although I do believe with proper explanations it should merit a higher score.

---

> ### Author Response · Authors · 2025-06-03
> **Response to Reviewer wVBW**
>
> We sincerely thank Reviewer wVBW for their thorough review and positive feedback, particularly on REFA's motivation, clarity of the paper, the URSLA insights, the solidity of gains, and our ablations. We appreciate this and have carefully considered all comments.
>
> We address the points raised:
>
> 1.  **Lack of Details on Method Variants:** This was a critical point. Our detailed rebuttal (see attached PDF, Rebuttal 1) provides clear definitions and loss functions for REFA-InfoNCA (and its "dynamic" sub-variant), REFA-1-vs-k (REFA-1vsk), and REFA-Dynamic. Specifically, as requested by Reviewer G1HG also, the content that is currently in Appendix E of our submission, will be moved to the main methodology in the revised paper to ensure clarity, as illustrated in the PDF. Specifically, in addressing this query, **we provide the content of the new subsection in our methodology for your perusal in the attached document below.**
>
> 2.  **Evaluation on Different Benchmarks & Pairwise Comparison:** We acknowledge the performance variations. AlpacaEval2, where REFA excels, strongly reflects benefits from length control and response richness. MT-Bench can show saturation.
>     *   **New On-Policy/Iterative Results (Table 1 in PDF):** Our new experiments demonstrate that in an on-policy/iterative setting, REFA variants achieve substantial improvements, particularly on challenging benchmarks. For example, on**Llama-Instruct (8B)**, REFA-Dynamic achieves **49.7% LC-WR** and **50.7% WR** on AlpacaEval2, and a strong **44.7% WR** on Arena-Hard. REFA-1vsk also performs very well with 49.6% LC-WR and 43.2% WR on Arena-Hard. In the iterative setting, REFA-Dynamic achieves  **52.17% LC-WR and 60.29% WR** on AlpacaEval2
>     *   **Pairwise Comparison:** We agree head-to-head comparisons are valuable. In the attached PDF (Rebuttal 2, Table 2), we present new pairwise win-rate improvements of REFA-Dynamic over SimPO on AlpacaEval2 in the on-policy setting, showing substantial gains (e.g., 60.6% for Llama-3-8b-Instruct win rate over SimPO). This provides a more direct measure of relative performance.
>
> 3.  **Different Trends for Llama-3 vs. Mistral:** Our detailed response (attached PDF, Rebuttal 3) explains that these differences likely stem from a combination of factors: architectural/pre-training differences, base model capabilities and initial SFT alignment, and the interaction between pre-training knowledge and how easily it's surfaced by different alignment signals for specific benchmarks like Arena-Hard.
>
> 4.  **Qualitative Examples:** This is an excellent suggestion. In the attached PDF (Rebuttal 4), we provide two initial qualitative examples comparing SimPO and REFA-Dynamic responses to illustrate REFA's benefits in generating more nuanced and contextually appropriate answers. We will add more such examples, including specific illustrations of EOS regularization's impact, to the appendix of the revised paper.
>
> We believe these clarifications, new results (especially the on-policy/iterative performance and pairwise comparisons), and planned revisions significantly strengthen our paper and address the reviewer's concerns. We have provided detailed responses and additional experiments in the attached document. Given these, we respectfully request Reviewer wVBW to reconsider their evaluation and would be **grateful if they would consider increasing their score**.
>
> [Link to Rebuttal_wVBW.pdf](https://drive.google.com/file/d/1ZQzcGsAwmcUTC_77NSbtr6D8dpUHYx8K/view)
>
> ---
>
> ### Summary of Important New Results:
>
> Our new experiments, detailed in the attached PDF, provide further strong evidence for REFA's capabilities:
>
> **Strong On-Policy/Iterative Performance:**
>
> REFA variants demonstrate significant leads over baselines on AlpacaEval2 and Arena-Hard for both Mistral-Instruct (7B) and Llama-Instruct (8B) in this setting. For instance, Llama-Instruct REFA-Dynamic achieves 49.7% LC-WR (AlpacaEval2) and 44.7% WR (Arena-Hard).
>
> **Significant Gains in Iterative Optimization:**
>
> When applied iteratively, REFA-Dynamic substantially outperforms a strong baseline like SPPO, achieving up to **52.17% LC-WR / 60.29% WR** on AlpacaEval2 and 52.70% WR on Arena-Hard for both Llama-3-8B and Mistral-7B SFT models after three iterations.
>
> **Direct Comparison & Statistical Significance:**
>
> Furthermore, new **direct pairwise comparisons** (detailed in the PDF) show REFA-Dynamic achieving substantial win-rate improvements (e.g., up to 60.6%) over strong baselines like SimPO in on-policy evaluations. Additionally, new **error analysis** (Std. Error and 95% CIs, also in the PDF) confirms the **statistical significance** of REFA's improvements in these on-policy/iterative settings.

---

> > ### Comment · Reviewer_wVBW · 2025-06-05
> >
> > Thanks for the very detailed and substantial responses. Given the additional experiments and promise for more methodological clarify in the revision, I've increased my assessment.

---

### Official Review · Reviewer_AyBr · 2025-05-19

**Rating:** 7
**Confidence:** 4
**Ethics Flag:** 1

**Summary:**

This paper proposes a family of reference-free alignment methods for multi-preference optimization which maintain response length control. The technical contributions that enable this are deviation-based weighting and a length regularizer. Theoretically, they show that naive length normalization can lead to shorter negative responses and shows how they can be addressed through additional constraints. Empirical results show REFA achieves state-of-the-art performance on AlpacaEval2, with 26.6% length-controlled and 24.2% standard win rate.

**Questions To Authors:**

1. Could you elaborate on the differences of different methods in terms of computational costs/efficiency?
2. From which tasks in the benchmarks does the improvement come from? It would be useful to provide an additional analysis to better understand the strengths.
3. The paper might benefit from an ablation study for the performance impact of each proposed component (deviation-based weighting, length normalization, EOS regularization).

**Reasons To Accept:**

- Introduces two new algorithmic components (deviation-based weighting) and improved length regularizer that addresses limitations in prior work.
- Provides some interesting theoretical insights into length-based optimization in alignment methods.
- Presents a thorough empirical comparison of the proposed methods with several competitive baselines across several datasets and different model families (up to 8B).
- Includes a detailed analysis the effect of length normalization hyperparameters and dynamic adaptation, highlighting useful insights for its use in practice. (e.g. need to carefully tune \lambda)

**Reasons To Reject:**

- The proposed methodology is incremental, primarily combining and refining techniques from existing approaches like SWEPO and SimPO.
- There is no formal proof in the theoretical foundation of URSLA, and it is mainly built with conjectures based on some empirical observations. It's unclear if the findings are universally valid.

---

> ### Author Response · Authors · 2025-06-03
> **Response to Reviewer AyBr**
>
> We sincerely thank Reviewer AyBr for their thorough review and for recognizing several key strengths of our work. We particularly appreciate the acknowledgment of REFA's novel algorithmic components, the interesting theoretical insights, our thorough empirical comparisons, and the detailed analysis of hyperparameter effects.
>
> We address the points raised for further clarification:
>
> 1.  **Incrementality of Methodology:** While REFA builds upon valuable concepts from prior work, its contributions are significant.
>
> As detailed in our attached PDF (Rebuttal 1, which includes a new subsection "Developing REFA: From Identified Needs to a Refined Loss"), REFA's key novelty lies in:
>
> A) **Theoretical Soundness:** Our principled integration of components like deviation-based weighting and length normalization with a specifically designed EOS-probability regularizer. This integration is uniquely motivated by our URSLA conjecture and its implications (Lemma 5), providing a strong theoretical rationale for achieving genuine fine-grained length control and richer outputs.
>
> B) **Systematic Development of Approach**: Furthermore, REFA-Dynamic is developed through a systematic process of addressing multiple identified weaknesses (W1-W5 from Appendix D) in simpler approaches. We outline this new section in the attached rebuttal pdf.
>
> C) **Stronger Results than the baselines mentioned** Finally, our new iterative results (summarized below and detailed in the PDF) also showcase REFA-Dynamic's distinct and strong performance, further distinguishing it from a mere combination of existing techniques.
>
> 2.  **Theoretical Foundation of URSLA:** We thank the reviewer for this point. URSLA (Conjecture 1) is presented as a formalization of an empirically observed and intuitively understood phenomenon regarding how model certainty evolves with sequence length.
>
> As detailed in our response (attached PDF, Rebuttal 2), the rationale is multi-faceted: as a model generates a response, it commits to a semantic trajectory, reducing uncertainty for subsequent tokens (supported by Figure 1 in our paper). This is particularly evident in multi-step reasoning (e.g., math, code generation) where initial ambiguity gives way to more predictable subsequent steps once a correct path is taken.
>
> This principle is also analogous to how extended coherent context generally aids prediction in sequence modeling (Beltagy et al., 2020). While not a universally proven law for all edge cases (which we acknowledge), URSLA provides a robust and insightful basis for understanding why naive length normalization can be insufficient and for motivating explicit length control mechanisms like our EOS regularizer, which demonstrably improve performance.
>
> 3.  **Computational Costs/Efficiency:** REFA is designed for efficiency.
>
> As shown in our detailed rebuttal (attached PDF, Rebuttal 3, Figure 1), when compared to a strong multi-preference baseline like SWEPO on Llama, REFA demonstrates a **17% reduction in training time (4h 1m for REFA vs. 4h 50m for SWEPO)** and a **10% decrease in peak GPU memory consumption (71GB for REFA vs. 79GB for SWEPO)**.
>
>
> 4.  **Ablation Study:** Our paper and its appendix already provide several analyses that function as ablations for REFA's key components, as detailed in our response (attached PDF, Rebuttal 4). These include:
>
>
> ####   **Deviation-Based Weighting:** The impact of the power $p$ in the deviation term is shown in Table 3 of the PDF (based on Table 2 of original submission), demonstrating benefits over an unweighted ($p=0$) version.
> ####  **EOS Regularization:** The effect of the EOS regularization strength $\lambda$ is directly ablated in Figure 2 of the PDF (based on Figure 2 of original submission), showing its crucial role in controlling length and optimizing win rates (with $\lambda=0$ being the no-EOS-regularization case).
> ####   **Negative Set Penalty $\gamma$:** Figure 3 of the PDF (based on Figure 3 of original submission) shows the impact of $\gamma$.
>
> We believe these clarifications, the demonstrated efficiency, existing ablations, and the supporting details in the attached PDF address the reviewer's points. Given this, and the strong performance (including new iterative results summarized below), we respectfully request Reviewer AyBr to reconsider their evaluation and would be grateful if they would consider increasing their score.
>
> [Link to Rebuttal_AyBr.pdf](https://drive.google.com/file/d/1XzZjdX0Cy5vVsixLH4k2D81PohG4Gxfh/view).
>
> ---
>
> We provide several new results in the rebuttal attached. For example:
>
> 1. **On-policy Results:** We show up to **52.17% LC-WR / 60.29% WR** on the iterative on-policy setting.
> 2. **Direct pairwise comparisons** between  REFA-Dynamic and SimPO show REFA achieving up to 60.6%  win-rate in on-policy evaluations.
> 3. **Error analysis (Std. Error and 95% CIs):** We provide error analyses which confirm the statistical significance of REFA's improvements in AlpacaEval2.0 and Arena-Hard.

---

> ### Comment · Area_Chair_XSEr · 2025-06-07
>
> Dear reviewer,
>
> thank you for your valuable effort. I noticed that you have not answered the authors rebuttal. Could please let us know what are your thoughts and if the answer has been satisfactory? Do you have additional questions?  Thank you!

---

> ### Author Response · Authors · 2025-06-10
> **Addressing Query on Efficiency via Different Methods (Regularizers)**
>
> We sincerely thank the reviewer for their feedback, which prompted us to conduct further experiments on token efficiency through  REFA's length control mechanisms and produce details on the resulting accuracy-efficiency tradeoffs here. We have prepared a detailed document with these new experiments, which we summarize as follows.
>
> ### Key Findings from New Experiments on Length Control (Full details in attached PDF):
>
> Our new experiments explore two types of EOS regularization within the REFA framework, performed on a ~20k subset of UltraFeedback.
>
> ### 1. Budget-Independent Length Control:
> Here the regularizer adapts to the length distribution of the training data itself.
>
>  **Effective Control & Clear Tradeoff:**
>
> We show that increasing the regularization strength ($\lambda$) effectively and predictably decreases the average response length (Figure 1 in PDF).
>
> **Optimal Performance Peak:**
>
> There is a clear performance peak for both Length-Controlled Win Rate (LC-WR) and raw Win Rate (WR) at a moderate $\lambda \approx 0.1$. This demonstrates that encouraging richer, longer responses improves alignment up to an optimal point, after which excessive verbosity becomes detrimental.
>
> **Systematic Distribution Shift:**
>
> Histograms of response lengths confirm that increasing $\lambda$ systematically shifts the entire generation distribution towards shorter responses, not just the average.
>
> ### 2. Fine-Grained Budgeted Length Control:
> For scenarios requiring precise control over token count, we designed a novel "budgeted" regularizer that penalizes termination before a target budget $b$ and incentivizes termination after.
>
> **Precise Budget Targeting:**
>
> Experiments with different budgets (e.g., $b=128, 256, 512$) show that this regularizer effectively steers the model's average output length to converge towards the target budget as regularization strength increases.
>
> **Tool for Efficiency-Accuracy Tradeoff:**
>
> This mechanism allows practitioners to explicitly set a token budget and observe the direct impact on alignment performance, providing a powerful tool for managing the accuracy vs. efficiency tradeoff based on specific application needs.
>
> **Insights on Fixed Budgets:**
>
> We observe that applying a single fixed budget to a dataset with diverse optimal response lengths can lead to complex (sometimes bi-modal) output distributions. This highlights that while budgeted control is powerful, adaptive strategies (like our primary budget-independent method) are crucial for datasets with high length variance.
>
> ---
>
> These experiments further demonstrate that REFA's EOS regularization provides a flexible and powerful framework for token efficiency through fine-grained length control. The full analysis, including all figures and detailed observations, is available in the attached PDF. We urge the reviewer consider our new experiments in response to their request on different methods for efficiency and would be **grateful if they would consider increasing their score.**
>
> [Link to Additional_Experiments.pdf](https://drive.google.com/file/d/1UDDvjmU0oiKuZxljMTPJthMEuKmcX03y/view)

---

### Official Review · Reviewer_G1HG · 2025-05-28

**Rating:** 6
**Confidence:** 3
**Ethics Flag:** 1

**Summary:**

This paper introduce REFA, which is new method for training language models to follow human preferences better. Previous methods make models give too short answers because they learn shortcuts, but users actually want longer and more detailed responses. REFA uses special techniques like length normalization and EOS probability control to fix this problem and make models generate richer, higher-quality answers. The experiments show that REFA performs much better than other methods on benchmarks like AlpacaEval2, achieving 26.6% Length-Controlled Win Rate compared to 21.5% for the previous best method SimPO.

**Questions To Authors:**

### Questions
* In Table 2, why is Arena Hard WR for SimPO (16.6) is not boldened but "REFA-dynamic (p=2)" (16.6) is boldened?
  * Also how are these hyper-parameters (`p`, $\alpha$, $\beta$, $\gamma$, $\lambda$) tuned? Is it using a separate development split for each dataset?

### Comments/Suggestions
* Appendix E is the key to this paper, and I strongly recommend shortening Section 3 and 4, then add (summarized version of) Appendix E after section 5. This is critical for understanding the flow of paper and made me very confused on the underlying motivation of proposing REFA and its variants.

**Reasons To Accept:**

* The paper contains with rich references and detailed technical background.
* Empirical results on uncertainty reduction Figure 1
* The paper has good motivation of proposing.
  * (Assuming the structure of this paper is massaged and Appendix E is correctly placed back in the main text)

**Reasons To Reject:**

* [minor] REFA needs careful hyperparameter tuning and variant selection (InfoNCA, 1-vs-k, or dynamic)
* Presentation and writing of the paper can be improved e.g., "REFA-dynamic" is the most important contribution of the work, but what exactly this refers to is hard to find.
   * Many key motivation for proposing REFA and variants are placed in Appendix E, which makes it hard to understand to overall sotry without reading Appendix E.
* The differences between other methods are none in some datasets (MT-Bench, Arena-Hard) and no error bars are present in the results.

---

> ### Author Response · Authors · 2025-06-03
> **Response to Reviewer G1HG**
>
> We sincerely thank Reviewer G1HG for their thorough review and for recognizing several strengths in our work, including our paper's rich references, detailed technical background, the empirical support for URSLA (Figure 1), and the strong motivation behind REFA. We appreciate this positive feedback.
>
> We address the points raised with clarifications and new supporting information:
>
> 1.  **Hyperparameter Tuning, Variant Selection, and Table Bolding (Weakness 1, Questions 1 & 2):**
>
> Our detailed rebuttal (see attached PDF, Rebuttal 1) clarifies our tuning strategy on a development split (UltraFeedback `test_prefs`) and highlights that many hyperparameters **exhibit consistent optimal values across models or settings** (e.g., $\alpha$ by model type, $\beta$ by model family, $\lambda$ for Instruct models), reducing the tuning burden. A full table of optimal hyperparameters used (Table 1 in the attached PDF) is provided.
>
>
> 2.  **Presentation, Motivation, Appendix E, and Clarity of REFA-Dynamic (Weaknesses 2 & 3, Comment/Suggestion 1):**
>
> This is a critical point, and we fully agree. The core logic and motivation for REFA variants (InfoNCA, 1-vs-k, and especially REFA-Dynamic), currently in Appendix E, will be integrated directly into the main methodology section. We have included a new subsection in the attached rebuttal PDF ("Developing REFA: From Identified Needs to a Refined Loss"). This subsection clearly defines the flow, making the paper's narrative and contributions much more accessible.
>
> 3.  **Relating to Error Bars (Weakness 4):**
>
> To address the request for error analysis and further demonstrate REFA's robustness, we have conducted **new experiments providing error analysis** (Std. Error and 95% CIs) for our REFA variants in an on-policy/iterative setting. These results (see attached PDF, Rebuttal 2, Tables 2 & 3) show that REFA's performance gains are statistically significant on AlpacaEval2 and Arena-Hard.
>
> Furthermore, our **new iterative training results** (Tables 4-6 in the "Additional Experimental Results" section of the attached PDF) showcase REFA-Dynamic achieving substantial gains on Arena-Hard (e.g., 52.70% WR for Llama-3-8B), highlighting stronger performance in this dynamic setting. We also briefly discuss benchmark-specific nuances for MT-Bench in the PDF.
>
>
> We believe these clarifications, the planned substantial revisions to the paper's structure, and the new supporting data (including error analysis and strong iterative results) address the reviewer's concerns. We have provided detailed responses and additional experiments in the attached document. Given these, we respectfully request Reviewer G1HG to reconsider their evaluation and would be **grateful if they would consider increasing their score** if our new experiments and responses have adequately addressed their queries.
>
> [Link to Rebuttal_G1HG.pdf](https://drive.google.com/file/d/14l9m7WQyCKQ8O06nd8dAhdFTDGxVz67W/view)
>
> ---
>
> ### Summary of Important New Results:
>
> Our new experiments, detailed in the attached PDF, provide further strong evidence for REFA's capabilities:
>
> **Strong On-Policy/Iterative Performance:**
>
> REFA variants demonstrate significant leads over baselines on AlpacaEval2 and Arena-Hard for both Mistral-Instruct (7B) and Llama-Instruct (8B) in this setting. For instance, Llama-Instruct REFA-Dynamic achieves 49.7% LC-WR (AlpacaEval2) and 44.7% WR (Arena-Hard).
>
> **Significant Gains in Iterative Optimization:**
>
> When applied iteratively, REFA-Dynamic substantially outperforms a strong baseline like SPPO, achieving up to **52.17% LC-WR / 60.29% WR** on AlpacaEval2 and 52.70% WR on Arena-Hard for both Llama-3-8B and Mistral-7B SFT models after three iterations.
>
> **Direct Comparison & Statistical Significance:**
>
> Furthermore, new **direct pairwise comparisons** (detailed in the PDF) show REFA-Dynamic achieving substantial win-rate improvements (e.g., up to 60.6%) over strong baselines like SimPO in on-policy evaluations. Additionally, new **error analysis** (Std. Error and 95% CIs, also in the PDF) confirms the **statistical significance** of REFA's improvements in these on-policy/iterative settings.

---

> > ### Comment · Reviewer_G1HG · 2025-06-07
> >
> > Thank you so much for the detailed responses.
> >
> > Looking at Table 1 in the attached PDF and Table 2 in the submitted draft, I believe it's still the case that when I want to apply it to either new model type, model family, or Instruct/non-Instruct models, then all parameters should be carefully tuned. But this is pretty common across preference (and general) training setup so no need to oversell REFA as "consistent" across settings (and I believe that's even misleading for the future readers). Marked as "[minor]" just to clarify that it's relatively less important.
> >
> > My core concerns for this paper is presentation and clarity in motivation and how does the new results fit with the original story of the paper, and to correctly evaluate on it, I will need a new revision/submission on it so I will keep my score as-is.

---

> > > ### Author Response · Authors · 2025-06-11
> > >
> > > Thank you again for the follow-up and the thoughtful clarification regarding the tuning concern. That makes complete sense — we agree that while there is some consistency in optimal hyper-parameters within certain families or settings, it wouldn’t be accurate to generalize that too broadly. We’ll make sure to revise the relevant phrasing in the paper to avoid overstating this point and to better reflect the practical tuning expectations for future users.
> > >
> > > On the core concern around clarity and presentation, we really appreciate you highlighting this. As mentioned in our response, we’ve already started reorganising the paper to bring the development of REFA variants and the motivation (currently in Appendix E) into the main methodology section. We’ve added a new subsection in the rebuttal PDF that walks through the design choices in a more cohesive narrative, and we’ll also make sure the latest results, especially the iterative setting experiments, are properly framed in the context of the original story.
> > >
> > > Your comments have been super helpful, and we’re confident the updated version will be much clearer and stronger thanks to your input. Thanks again!
> > >
> > > [Link to Rebuttal_G1HG.pdf](https://drive.google.com/file/d/14l9m7WQyCKQ8O06nd8dAhdFTDGxVz67W/view?usp=sharing)

---

> ### Comment · Area_Chair_XSEr · 2025-06-07
>
> Dear reviewer,
>
> thank you for your valuable effort. I noticed that you have not answered the authors rebuttal. Could please let us know what are your thoughts and if the answer has been satisfactory? Do you have additional questions?  Thank you!

---

### Author Response · Authors · 2025-06-10
**Acknowledging Reviews and Strengths of the paper.**

We are deeply grateful to all reviewers for their time and insightful feedback. We were particularly encouraged by the positive recognition of several aspects of our work across the reviews, including:

- REFA's motivation and its interesting approach to controlling dataset biases (Reviewers G1HG, 9iEL, wVBW, zZqs).
- The novelty of algorithmic components like deviation-based weighting and the improved length regularizer (Reviewer AyBr).
- The interesting theoretical insights, especially regarding URSLA and length-based optimization (Reviewers AyBr, wVBW).
- The paper's rich references and detailed technical background (Reviewer G1HG).
- Our clean and thorough experimental setup and comparisons (Reviewers 9iEL, AyBr, zZqs).
- The solidity of our empirical results and good performance on standard benchmarks (Reviewers wVBW, zZqs).
- The value of our ablation studies and hyperparameter analyses (Reviewers AyBr, wVBW).


We appreciate this positive acknowledgment and have aimed to build upon these strengths while addressing all concerns raised.

---

> ### Author Response · Authors · 2025-06-10
> **Summary of New Experiments and Key Findings**
>
> In response to the reviewer feedback, we conducted additional experiments to further demonstrate REFA's capabilities in on-policy settings and for fine-grained length control for token efficiency. These new results underscore the robustness and versatility of our method. (Full details, tables, and figures are in the attached PDF).
>
> ## 1. Strong Performance in On-Policy and Iterative Settings
>
> To address questions about performance beyond static off-policy data, we evaluated REFA in settings that more closely approximate on-policy learning.
>
> **Improved On-Policy Performance**
>
> When applied in an on-policy setting (generating responses from the model being trained), REFA variants show significant leads over baselines. For instance, on Llama-Instruct (8B), REFA-Dynamic achieves **49.7% LC-WR** on AlpacaEval2 and **44.7% WR** on Arena-Hard (Table 1 in PDF).
>
> **Even better with Increasing Number of Iterations**
>
> When applied iteratively, REFA-Dynamic substantially outperforms a strong iterative baseline (SPPO). After three iterations, REFA-Dynamic achieves up to **52.17% LC-WR / 60.29% WR** on AlpacaEval2 and **52.70% WR** on Arena-Hard, demonstrating its effectiveness in progressive model refinement (Tables 3 & 4 in PDF).
>
> ## 2. Fine-Grained Length Control via EOS Regularization
>
> We provide a detailed analysis of our two EOS regularization mechanisms, showing how they provide precise control over output length and manage the accuracy-efficiency tradeoff.
>
> **Budget-Independent Control**
>
> Our primary regularizer (Equation 1 in PDF) demonstrates that by adjusting a single parameter, $\lambda$, we can effectively control average response length. As shown in Figure 1 of the PDF, a moderate $\lambda \approx 0.1$ improves both LC-WR and WR, while higher values can be used to systematically reduce length to meet efficiency needs.
>
> **Budgeted Control for Specific Length Targets**
>
> A novel "budgeted" regularizer (Equation 2 in PDF) allows practitioners to target a specific token budget $b$. Experiments (Figure 2 in PDF for $b=256$) show this provides a powerful tool for enforcing length constraints while observing the direct impact on alignment quality.
>
> These new results confirm that REFA is not only a strong offline alignment method but also highly effective and controllable in more dynamic, on-policy, and iterative settings. We believe this significantly strengthens the contributions of our work.
>
> ---
>
> Full details, including all tables and figures for these new experiments, are available in our consolidated response document:
>
> [Link to Additional Experiments](https://drive.google.com/file/d/1lu2rIPdEjvcLaZD_RevcQoULxhyFFwnE/view)

---

### Decision · Program_Chairs · 2025-07-08

**Decision:**

Accept

**Comment:**

This paper introduces REFA, a new method for training language models to better follow human preferences by addressing optimization shortcuts that can lead to overly short answers. REFA uses a combination of deviation-based weighting to focus on high-impact examples, length normalization, and a specific End-of-Sequence (EOS) probability regularizer to encourage richer, higher-quality outputs. Experiments show that this approach allows REFA to perform significantly better than other reference-free methods on benchmarks such as AlpacaEval2.

Based on the reviewers' observations and evaluation, and given the positive discussion and clarifications during the review period, I agree that the paper: i) the paper is well structured and motivated; ii) the experimental setup is sound iii) the findings and proposed methods are exciting and relevant for the community; iv) the empirical results support the motivation and theoretical claims. The risks and weaknesses were addressed, clarified, or can be included in the final version of the paper.

Therefore, I recommend this paper to be accepted. I would suggest that the authors include the complementary results and incorporate the discussion and  recommendations provided by the reviewers.